# KIF11 prevents retinal endothelial ferroptosis in familial exudative vitreoretinopathy by inhibiting phosphorylation-driven PRDX1 phase separation

Mu Yang [1,2,6], Rulian Zhao[3,6], Li Peng [1,6], Liting Lv [1,6], Lanyao Yang [1], Yunqi He [1], Xu Ha [1], Huijuan Xu [1], Xiang Zhang [4], Peiquan Zhao [4], Shujin Li [1,2,7] ✉ & Zhenglin Yang [1,2,5,7] ✉

Familial exudative vitreoretinopathy is a hereditary disorder predominantly affecting infants and young children, often leading to severe vision loss. Approximately 40% of patients carry mutations in Norrin/β-catenin pathway genes. Nevertheless, the downstream pathogenic mechanisms remain unclear. Here, by using bulk RNA sequencing and single-cell RNA sequencing analyses, we identify *KIF11* as a key downstream effector in retinal endothelial cells. Lentivirus-mediated KIF11 overexpression partially restores vascular defects in endothelial cell-specific *Ctnnb1* knockout mice. Functional and multi-omics studies reveal that β-catenin/KIF11 deficiency induces autophagy-accompanied ferroptosis. Mechanistically, KIF11 binds PRDX1, and the disrupted β-catenin/KIF11 axis releases the competitive restraint of KIF11 on Src-mediated PRDX1 phosphorylation, triggering subsequent liquid-liquid phase separation. Treatment with the ferroptosis inhibitor ferrostatin-1 or lentiviral overexpression of non-phosphorylatable PRDX1 partially rescues vascular defects in familial exudative vitreoretinopathy-associated mice. Overall, we elucidate a β-catenin/KIF11/PRDX1 axis-dependent ferroptosis mechanism in familial exudative vitreoretinopathy, highlighting ferroptosis-targeting and antioxidant strategies as potential therapies.

Familial exudative vitreoretinopathy (FEVR) is a hereditary eye disease (0.46% in Chinese newborns) driven primarily by mutations in the Norrin/β-catenin (~40% of cases, *FZD4*, *LRP5*, *NDP*, *TSPAN12*, or *CTNNB1*)[1,2], as well as the cadherin/catenin (*CTNNA1* and *CTNND1*)[3,4] and Jag1/Notch signaling (*JAG1*)[5]. Despite the central role of Norrin/β-catenin signaling, the key downstream target of Norrin/β-catenin is unknown, hindering the development of efficient therapies for FEVR.

KIF11 was initially identified as a kinesin motor protein with end-directed motor activity and is known to associate with spindle microtubules[6]. Overexpression of KIF11 has been associated with solid

tumors and poor prognosis[7], whereas its deficiency prevents the separation of duplicated centrosomes, leading to the formation of monoastral spindles and subsequent mitotic block[8]. *KIF11* mutations are associated with FEVR, and a recent study demonstrated that endothelial cell (EC)-specific knockout of *Kif11* recapitulated FEVR-like phenotypes[9]. However, in contrast to most FEVR-associated genes, the regulatory role of KIF11 in Norrin/β-catenin signaling remains unclear.

Ferroptosis is a form of programmed cell death characterized by the accumulation of cellular iron and lipid peroxidation[10]. Beyond its involvement in pathological conditions like cancer, sepsis, and

degenerative diseases, emerging evidence indicates that ferroptosis is associated with vascular damage like pulmonary arterial hypertension (PAH), atherosclerosis, and lung fibrosis[11–13]. However, whether the Norrin/β-catenin signaling or KIF11 exerts a similar effect against ferroptosis in FEVR remains unresolved. Our present study aimed to elucidate the pathogenic mechanisms downstream of decreased β-catenin signaling. To achieve this, we conducted a combination of multi-omics analysis with cell-based models and animal models to reveal that KIF11 is a pivotal downstream regulator of the β-catenin signaling pathway. We also demonstrate that loss of KIF11 function in endothelial cells led to autophagy-accompanied ferroptosis by releasing its competitive role on the Src-mediated phosphorylation and triggering liquid-liquid phase separation (LLPS) of PRDX1, ultimately causing defective vascularization in FEVR.

## Results

### KIF11 is a key downstream target of Norrin/β-catenin signaling

To identify genes strongly associated with reduced Norrin/β-catenin signaling, we performed unbiased bulk RNA sequencing (RNA-seq) to analyze the global transcriptional landscape of human retinal microvascular endothelial cells (HRECs) following knockdown of genes in the Norrin/β-catenin signaling pathway, specifically CTNNB1, FZD4, TSPAN12, or LRP5 (Fig. 1a, b and Supplementary Data 1). Depletion of these genes markedly altered the expression of representative β-catenin downstream targets[14–16], characterized by downregulation of APCDD1, DKK1, and SLC2A1, along with the upregulation of PLVAP (Fig. 1c). Dimension reduction via 3D principal coordinate analysis (PCoA) revealed convergent transcriptional responses of the CTNNB1, FZD4, TSPAN12, and LRP5 knockdown cells (Supplementary Fig. 1a). Moreover, Venn analysis showed that approximately 25% of differentially expressed genes (DEGs) in each group were shared among all four knockdown groups, indicating a common transcriptomic response to Norrin/β-catenin pathway inhibition (Supplementary Fig. 1b–d and Supplementary Data 2). Notably, KIF11 was among the top 76 downregulated genes across the knockdown conditions (Supplementary Data 2).

Weighted coexpression network analysis (WGCNA) identified 13 gene modules, among which the blue and dark gray modules were selected for further analysis owing to their strongest and most consistent associations with knockdown-induced transcriptomic changes (Fig. 1d, e). The blue module showed a robust negative correlation with the control group (ME = −0.71, p = 0.003) and uniformly positive correlations across four knockdown groups, compared to the opposite regulation pattern in the dark gray module (ME = 0.98, $p = 3e^{-10}$) (Fig. 1d, e). Kyoto Encyclopedia of Genes and Genomes (KEGG) pathway analysis showed significant enrichment of blue module genes in cell-death pathways, especially ferroptosis (Fig. 1f).

As β-catenin activates transcription with TCF/LEF proteins[17], genes within the dark gray module were regarded as the primary targets regulated by the loss of β-catenin signaling. KEGG pathway analysis revealed the enrichment of these genes with cell cycle and meiosis (Fig. 1g), which aligns with the established link between compromised β-catenin signaling and cell proliferation[18]. PPI network analysis of genes in the dark gray module using CytoHubba maximal clique centrality (MCC) algorithm revealed KIF11, a known FEVR-associated gene, as one of the top five-ranked downstream targets of Norrin/β-catenin signaling (Fig. 1h, i). Quantitative real-time PCR (RT-qPCR) and western blot consistently confirmed a robust downregulation of KIF11 following CTNNB1, FZD4, TSPAN12, or LRP5 knockdown (Fig. 1j–q and Supplementary Fig. 1e–h). Furthermore, exogenous Norrin increased KIF11 expression in HRECs (Supplementary Fig. 1i). Consistently, KIF11 remained significantly downregulated upon knockdown of these genes (Supplementary Fig. 1j). These results suggest KIF11 as a potential core target of Norrin/β-catenin signaling.

### Kif11 is expressed in proliferative mouse retinal ECs and transcriptionally regulated by Norrin/β-catenin signaling

We next sought to delineate the cellular expression landscape of Kif11 in postnatal and adult mouse retinal ECs with or without Norrin/β-catenin inactivation. We constructed EC-specific Ctnnb1 conditional knockout (Ctnnb1flox/flox; Pdgfb-iCreER[19], Ctnnb1 cKO) and Tspan12 knockout (Tspan12 KO) mice, which showed efficient gene deletion and characteristic FEVR-like retinal vascular defects (Fig. 2a and Supplementary Fig. 2a, b). We next performed single-cell RNA sequencing (scRNA-seq) on retinal vascular cells isolated from postnatal day 6 (P6) wild-type (WT), Ctnnb1 cKO, and Tspan12 KO mice, and integrated these data with publicly available datasets from P26 WT and Tspan12 KO retinas (GSE213887)[16], P60 WT and P60 Ndp KO retinas (GSE125708)[20], as well as ECs of P6 WT retinas (GSE175895)[21] (Fig. 2b). After stringent quality control, a total of 78,830 retinal cells were annotated based on published markers[20], including 5338 ECs (Fig. 2c and Supplementary Fig. 2c).

Focusing on ECs, we observed consistent downregulation of Apcdd1, and upregulation of Plvap in the cKO or KO groups compared with their corresponding WT, confirming pathway suppression upon gene ablation (Fig. 2d). Kif11 expression in ECs of postnatal (P6) mice, rather than in adult (P26, and P60) mice, were downregulated upon inactivation of Norrin/β-catenin signaling (Fig. 2d). Notably, Kif11 was predominantly expressed in postnatal retinas compared with adult retinas (Fig. 2d and Supplementary Fig. 2d), suggesting a critical role for KIF11 during retinal development. Further reclustering of the ECs identified seven EC subpopulations based on previously established arterio-venous (A-V) zonation markers[22], including arterial capillary EC, arterial EC, capillary EC, tip EC, proliferative EC, venous capillary EC, and venous EC (Fig. 2e and Supplementary Fig. 2e, f). As expected, proliferative ECs were exclusively derived from postnatal retinas, where Kif11 was selectively enriched and downregulated upon inactivation of Norrin/β-catenin (Fig. 2f–h and Supplementary Fig. 2e). Therefore, KIF11 is likely expressed during postnatal retinal development, as validated by RT-qPCR and western blot analyses of retinas collected from P1, P7, P14, P21, and P28 (Fig. 2i and Supplementary Fig. 2g).

We next turned to a publicly available transposase-accessible chromatin using sequencing (ATAC-seq) dataset of cerebellar endothelial cells isolated from control mice and mice with endothelial cell-specific β-catenin stabilization[23]. β-catenin stabilization led to a genome-wide increase in chromatin accessibility at transcription start sites (TSSs), with increased chromatin accessibility at the promoter region of Kif11, as well as Axin2 (Fig. 2j, k and Supplementary Fig. 2h), a known β-catenin downstream target[23]. Further analysis of chromatin immunoprecipitation sequencing (ChIP-seq) data for TCF7L2[24], a major cofactor and transcriptional effector of Norrin/β-catenin signaling pathway, revealed evident binding of TCF7L2 near the TSSs of both Kif11 and Axin2 (Fig. 2l–n). Collectively, these results reveal that Kif11 is specifically expressed in proliferative mouse retinal ECs and transcriptionally downregulated upon impaired β-catenin signaling during vascular development.

### Pathogenic KIF11 mutations and functional rescue in FEVR

In previous studies, mutations in the KIF11 gene were shown to be associated with FEVR[25]. We applied targeted gene panel on the genomic DNA samples from FEVR patients and identified six predicted deleterious KIF11 mutations, including NM_004523.4: c.466_469del (p.Val156LysfsTer38), c.1387dupA (p.Thr463AsnfsTer10), c.2342delA (p.Gln781ArgfsTer18), c.2579C > A (p.Ser860Ter), c.2738del (p.Leu913ArgfsTer5), and c.2906_2907del (p.Lys969ArgfsTer19) (Supplementary Fig. 3a and Supplementary Table 1). Clinically, probands in families 1, 3, and 5 experienced exudative retinal detachment of both eyes, while the probands in families 4 and 6 exhibited retinal proliferative lesions accompanied by localized tractional retinal

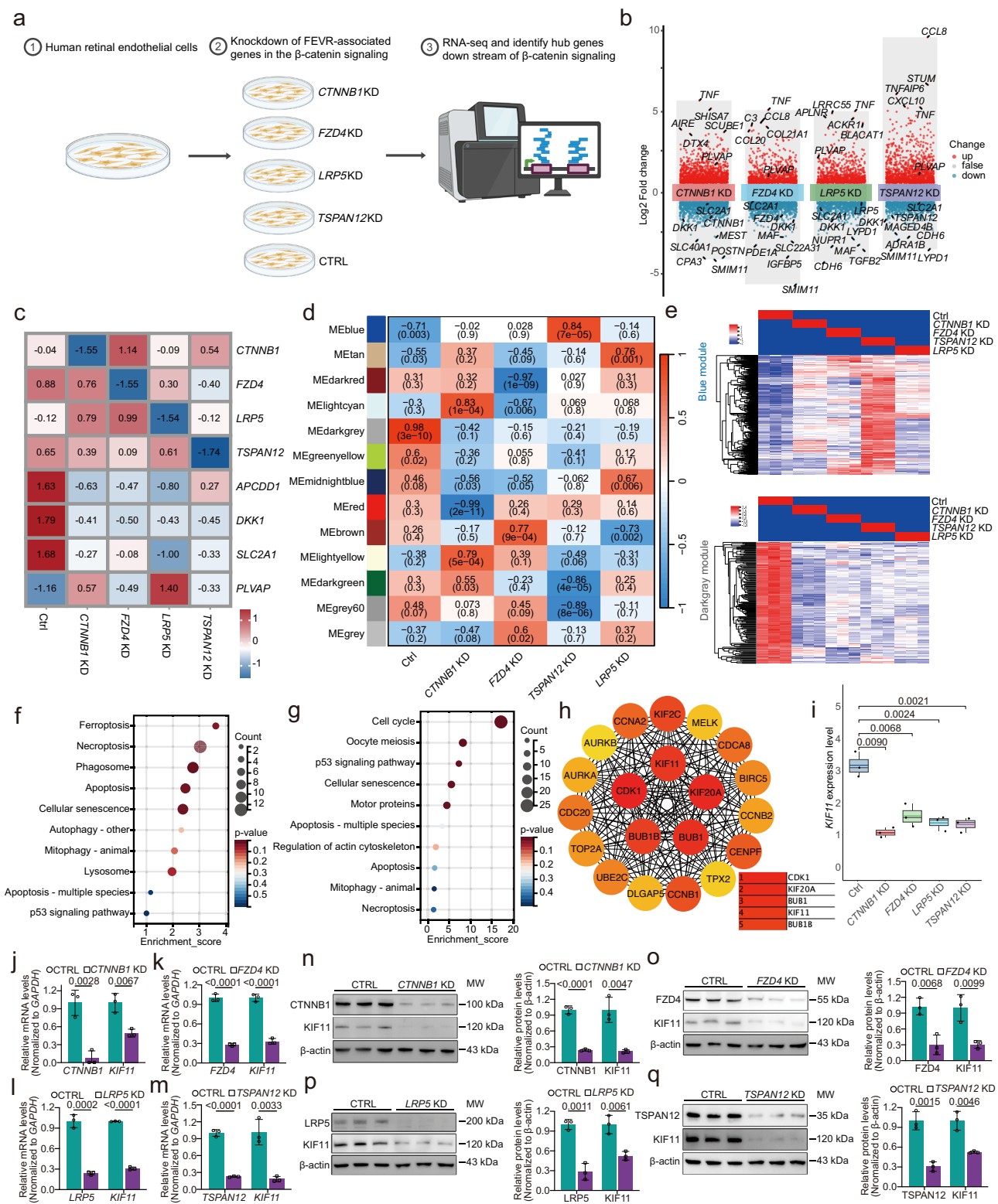

detachment in both eyes (Supplementary Fig. 3b–i). Western blot analysis revealed truncation and moderate degradation of these overexpressed mutant proteins in HEK293T cells (Fig. 2o), which may underlie the pathogenesis of FEVR.

We next asked whether restoring KIF11 could rescue the retinal vascular defects in *Ctnnb1* cKO mice. Retro-orbital venous sinus delivery of lentivirus exhibited efficient and targeted transduction in retinal vascular cells, as revealed by immunofluorescence of GFP (Supplementary Fig. 3j). Remarkably, lentiviral overexpression of *KIF11*

via retro-orbital venous sinus injection exhibited partial restoration of retinal vascular progression in *Ctnnb1* cKO mice without affecting their body weight (Fig. 2p–r and Supplementary Fig. 3k). Collectively, these results provide direct evidence that KIF11 functions as a critical downstream effector of Norrin/β-catenin signaling. Downregulation of KIF11 due to impaired Norrin/β-catenin signaling, or *KIF11* loss-of-function mutations leading to protein truncation and degradation, may represent a primary mechanism underlying FEVR.

**Fig. 1 | Bulk RNA-seq reveals KIF11 as a top-ranked hub downstream target of Norrin/β-catenin signaling pathway. a** Schematic diagram of the integrative bulk RNA-seq analysis in HRECs depleted with *CTNNB1*, *FZD4*, *LRP5*, or *TSPAN12*, using the same batch of control (CTRL) cells. Created in BioRender. Li, S. (2026) https://BioRender.com/b75j366. **b** Differential gene expression analysis showing the upregulated (red) and downregulated (blue) genes in *CTNNB1*, *FZD4*, *LRP5*, or *TSPAN12* knockdown (KD) HRECs. **c** Heatmap of representative β-catenin downstream targets and knockdown genes. **d** Heatmap of WGCNA module-trait relationships, where rows represent modules and columns represent HRECs transduced with CTRL, *CTNNB1*, *FZD4*, *LRP5*, or *TSPAN12* shRNA. **e** Heatmap of genes from blue and dark gray modules across all samples. KEGG pathway enrichment of biological process signalings for genes within blue (**f**) and dark gray (**g**) modules. **h** Hub genes identified by Cytohubba plug-in (MCC, Maximal Clique Centrality) within the dark gray module. **i** Differential expression of *KIF11* in CTRL versus *CTNNB1* KD, *FZD4* KD, *LRP5* KD, or *TSPAN12* KD HRECs by bulk RNA-seq, with Log$_2$FC of −1.62, −0.92, −1.22, and −1.23, respectively. Box plots show the median (center line), 25th–75th percentiles (bounds of box), and whiskers extending to the minima and maxima values. Relative mRNA levels of *KIF11*, *CTNNB1*, *FZD4*, *LRP5*, and *TSPAN12* in CTRL versus *CTNNB1* KD (**j**), *FZD4* KD (**k**), *LRP5* KD (**l**), or *TSPAN12* KD (**m**) HRECs. *n* = 3. Western blot analyses and relative quantification of protein levels of KIF11 together with CTNNB1, FZD4, LRP5, or TSPAN12 in CTRL versus *CTNNB1* KD (**n**) *FZD4* KD (**o**), *LRP5* KD (**p**) or *TSPAN12* KD (**q**) HRECs, respectively. *n* = 3. Data are presented as mean ± SD. *n* represents independent biological replicates. Statistical significance was determined using a two-tailed Student's *t* test (**i–q**). Source data are provided as a Source Data file. MW molecular weight.

## KIF11 deficiency leads to decreased proliferation and increased HREC death

To explore how KIF11 dysfunction contributes to FEVR, we knocked down *KIF11* in HRECs using an shRNA-expressing lentivirus (Supplementary Fig.4a–c). *KIF11* depletion inhibited proliferation and triggered cell death, as revealed by real-time cellular analysis (RTCA) (Fig. 3a) and EdU incorporation assays (Supplementary Fig. 4d, e). While TUNEL assays detected no overt apoptosis in *KIF11*-knockdown HRECs (Supplementary Fig. 4f), transmission electron microscopy (TEM) revealed that *KIF11* depletion induced increased autophagosome formation and ferroptosis-like mitochondrial alterations, including cristae loss and outer membrane fragmentation (Fig. 3b). Thus, the alterations observed in *KIF11* knockdown HRECs are strongly related to the molecular pathways perturbed by reduced β-catenin signaling (Fig. 1f, g).

To further investigate the molecular consequences of KIF11 deficiency, we performed bulk RNA-seq (Supplementary Data 1). Differential expression and KEGG analyses indicated that genes downregulated upon *KIF11* knockdown were predominantly involved in the cellular process of cell cycle (Supplementary Fig. 4g, h). Gene set enrichment analysis (GSEA) revealed that KIF11-deficient cells were negatively correlated with cell cycle and DNA replication pathways (Supplementary Fig. 4i–o), recapitulating the transcriptional effects of compromised β-catenin signaling (Fig. 1g). Moreover, several motor proteins were also downregulated, highlighting a regulatory role for KIF11 in motor activity (Supplementary Fig. 4n, o). Conversely, genes upregulated upon *KIF11* knockdown were enriched in cellular processes related to necroptosis, phagosome, and lysosome signaling pathways but were less associated with ferroptosis (Supplementary Fig. 5a). Consistently, GSEA revealed robust enrichment of lysosome and autophagy pathways, in contrast to ferroptosis (Supplementary Fig. 5b–f). RT-qPCR further validated increased expression of autophagy-related genes (*ATG5*, *ATG7*, and *ATG14*), and core components of autophagic machinery (*BECN1*, *MAP1LC3B*, *TFEB*, and *CLEAR*), as well as lysosomal marker *LAMP1* (Supplementary Fig. 5g), confirming activation of autophagy and lysosome signaling in KIF11-deficient HRECs.

## Integrated analysis revealed the involvement of KIF11 in autophagy-accompanied ferroptosis

We next used mass spectrometry to profile proteomic changes upon *KIF11* knockdown. A total of 555 proteins were upregulated, and 538 were downregulated in the absence of KIF11 (Fig. 3c). GSEA and western blot validation revealed strong negative correlations between KIF11-deficient cells and cell cycle, DNA replication, and motor protein signaling (Supplementary Fig. 6a–h), consistent with the transcriptomic data. In contrast, KEGG pathway analysis of all differentially expressed proteins identified ferroptosis, peroxisome, phagosome, and lysosome as the top-enriched pathways, while the cycle and DNA replication were less significantly enriched (Fig. 3d). Accordingly, GSEA revealed strong positive correlations between *KIF11*-depleted cells and

lysosome, autophagy, ferroptosis, and peroxisome pathways (Fig. 3e and Supplementary Fig. 6i–n), indicating increased autophagy and ferroptosis-like alterations upon *KIF11* depletion. We then integrated the transcriptomic and proteomic data using a nine-quadrant diagram (Fig. 3f) and found that most of the altered genes (in quadrants four and six) exhibited significant expression changes at the protein level but minimal changes at mRNA level (Fig. 3g). KEGG pathway analysis of genes with unchanged transcription but altered protein levels (i.e., those in quadrants four and six) was largely consistent with those shown in Fig. 3d (Supplementary Fig. 7a). These findings not only confirm the role of KIF11 in regulating cell proliferation but also suggest that KIF11 participates in regulating cell death-related pathways, particularly ferroptosis, mainly at the protein level.

We then first investigated the regulatory role of KIF11 in autophagy. Western blot analysis revealed that KIF11 deficiency increased the expression of core autophagy- and lysosome-related proteins, including ATG5/7/14, Beclin-1, and LAMP1, in addition to decreasing SQSTM1/p62 levels and increasing the conversion of LC3I to LC3II, strongly indicating enhanced autophagic flux (Supplementary Fig. 7b, c). Immunofluorescence further confirmed a marked increase in the number of LC3B puncta and elevated LAMP1 expression in *KIF11* knockdown cells (Supplementary Fig. 7d–f). Taken together, these results suggest that the loss of KIF11 might trigger autophagy in HRECs.

Ferroptosis requires selective autophagic degradation of anti-ferroptotic proteins such as glutathione peroxidase (GPX) 4, thereby facilitating the generation of ROS and lipid peroxides[26]. Consistent with this, transcriptomic and proteomic analyses revealed a pronounced decrease in GPX4 protein levels with minimal changes in mRNA following *KIF11* knockdown (Fig. 3h). RT-qPCR further revealed considerable increases in the transcription of the ferroptosis sensitivity marker *ACSL4*[27] and ferroptosis regulatory factor *HMOX1* (*HO-1*)[28], while most other ferroptosis-related genes showed minimal alterations at mRNA level (Supplementary Fig. 7g). Conversely, the protein levels of xCT, PCBP1, GPX4, and SLC3A2 were downregulated, while ACSL4 and HO-1 were upregulated (Fig.3i, j). Additionally, the protein levels of the ferroptosis regulator, ferroptosis suppressor protein 1 (FSP1)[29] and dihydroorotate dehydrogenase (DHODH)[30], remained largely unchanged (Fig. 3i, j), indicating that KIF11 may selectively regulate ferroptosis via the xCT/GSH/GPX4 axis. Consistently, the alterations of the core ferroptosis-related proteins were recapitulated with another *KIF11* knockdown shRNA (Supplementary Fig. 7h–j).

Glutathione (GSH) is thought to be the primary coordinating ligand for cytosolic ferrous iron and serves as the substrate for GPX4[31]. The enzymatic activity of GPx was measured using a kinetic spectrophotometric assay based on the reaction between GSH and 5,5′-dithio-bis-(2-nitrobenzoic acid) (DTNB)[32], which revealed an approximately 50% decrease in GPx activity following *KIF11* knockdown (Fig. 3k). Consistently, knockdown of *KIF11* significantly increased the cellular level of labile ferrous iron (Fig. 3l, m). Ferroptosis is characterized by the accumulation of ROS and lipid peroxidation[11,33]. In line with this, the levels of malondialdehyde (MDA), BODIPY™ 665/676, and CellROX

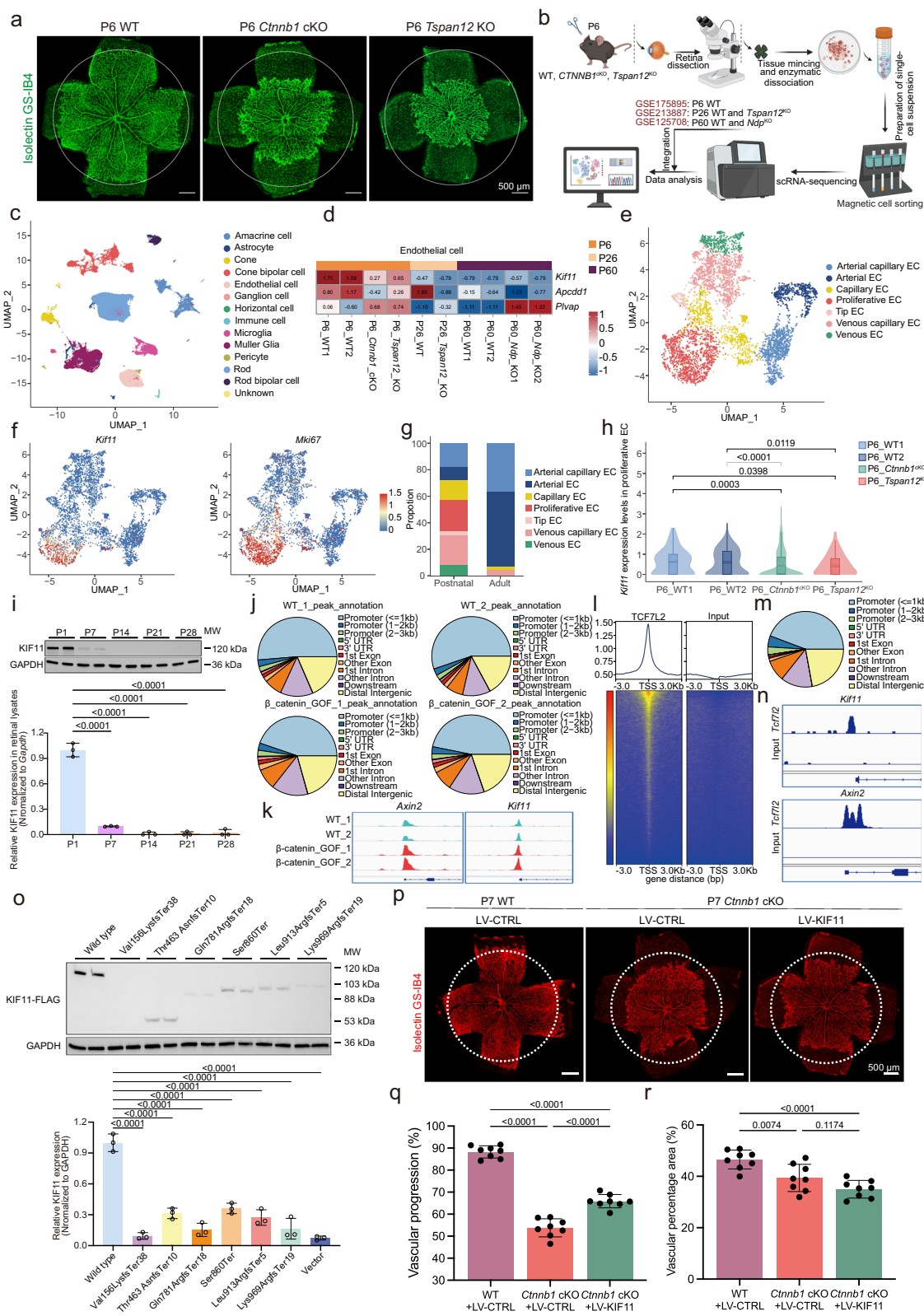

were markedly elevated in *KIF11*-knockdown HRECs (Fig. 3n–r). Liquid chromatograph-tandem mass spectrometer (LC-MS/MS) targeted analysis demonstrated that *KIF11* depletion in HRECs enhanced the peroxidation of phosphatidylethanolamines (PEs), phosphatidylcholines (PCs), and their ether-linked forms (Fig. 3s), which are associated with ferroptosis[11,33]. Time-lapse real-time live-cell imaging using Annexin V and propidium iodide (PI) showed increased cell death upon

*KIF11* knockdown (Supplementary Fig. 8a–c), which was corroborated by elevated lactate dehydrogenase (LDH) release (Supplementary Fig. 8d). These results imply that *KIF11* depletion increases ferroptotic vulnerability in HRECs.

We next asked whether treatment with the ferroptosis inhibitor ferrostatin-1 (Fer-1) could restore endothelial functions. Fer-1 treatment effectively rescued the ferroptotic phenotypes induced by *KIF11*

**Fig. 2 | The expression and role of KIF11 in FEVR. a** Representative images of retinal flat mounts from P6 wild-type (WT), *Ctnnb1* conditional knockout (cKO) mice, and *Tspan12* knockout (KO) mice. Circles indicate vessel outgrowth of the control. Green, Isolectin GS-IB4. Scale bars, 500 μm. **b** Schematic diagram of scRNA-seq from isolated retinal vascular cells and publicly available datasets. Created in BioRender. Li, S. (2026) https://BioRender.com/b75j366. **c** UMAP diagram showing distinct cell clusters of retinas. **d** Heatmap of the relative *Kif11*, *Apcdd1*, and *Plvap* expression within endothelial cells (ECs) across samples. **e** UMAP diagram of EC subpopulations. **f** UMAP plot of *Kif11* and *Mki67* expression. **g** Stacked bar plot depicting the composition of endothelial cell subpopulations in postnatal and adult groups. **h** Violin plots showing *Kif11* expression in proliferative ECs. The outer violin outline represents the kernel probability density. The embedded box plots indicate the median (center line) and mean (dashed line), the 25th and 75th percentiles (bounds of the box), and the whiskers extending to the minima and maxima. *n* = 134, 355, 491, and 66 cells per group. **i** Western blot and relative quantification of KIF11 in retinas from different time points. *n* = 3. **j** Pie charts showing annotation of ATAC-seq peak distributions. **k** ATAC-seq peak signals of *Kif11* and *Axin2* in isolated cerebellar ECs from control and EC-specific β-catenin-stabilized mice. **l** Heatmap of ChIP-seq signal intensity around transcription start sites (TSS). **m** Genome-wide distribution of ChIP-seq peaks. **n** ChIP-seq peak signals of *Kif11* and *Axin2*. **o** Western blot and relative quantification of WT and mutant KIF11 protein levels. *n* = 3. Representative images (**p**) and quantification of retinal vascular progression (**q**) and vessel density (**r**) from WT and P7 *Ctnnb1* cKO mice treated with CTRL or KIF11-overexpressing lentivirus. Circles indicate vessel outgrowth of the KIF11-overexpressed mice. Red, Isolectin GS-IB4. Scale bars, 500 μm. *n* = 8. Data are presented as mean ± SD. *n* represents independent biological replicates or the number of mice per group. Statistical significance was determined using two-tailed Student's *t* test (**h**), one-way ANOVA with Dunnett's (**i**, **o**), or Tukey's (**q**, **r**) multiple comparisons test. Source data are provided as a Source Data file.

depletion in HRECs, including impaired viability, lipid peroxide accumulation, elevated ROS and labile iron, reduced GPx activity, and dysregulated ferroptosis-related proteins (Fig. 4a–m). To assess whether KIF11 plays a broader role in ferroptosis beyond endothelial cells, we conducted additional experiments in HT1080 cells, a model system commonly used for ferroptosis studies[10]. As expected, *KIF11*-depleted cells exhibited enhanced ferroptosis, with reduced xCT, PCBP1, GPX4, and SLC3A2 expression, increased lipid peroxidation, and elevated cell death (Supplementary Fig. 8e–k). Fer-1 treatment also effectively rescued the ferroptotic phenotypes in *KIF11*-deficient HT1080 cells (Supplementary Fig. 8l, m). Collectively, these data demonstrate that the absence of *KIF11* triggers autophagy-accompanied ferroptosis.

### KIF11 interacts with PRDX1 to suppress the LLPS of PRDX1

We next performed immunoprecipitation-mass spectrometry in KIF11-FLAG-overexpressing HRECs to examine whether KIF11 protects against ferroptosis via protein interactions (Fig. 4n). The mass spectrometry identified nine top-ranked KIF11 binding partners (Fig. 4o), among which IK and PRDX1 remained unchanged following *KIF11* knockdown or overexpression (Supplementary Fig. 9a–c). Nevertheless, Co-IP confirmed that PRDX1, but not IK, interacts with KIF11 (Fig. 4p). Importantly, FEVR-associated truncated *KIF11* variants nearly eliminated its interaction with PRDX1 (Fig. 4q).

Since PRDX1 is a thiol-dependent peroxidase with a conserved cysteine (Cys) residue that serves as the site for peroxide-mediated oxidation[34], we hypothesized that KIF11 loss induces ferroptosis via inhibition of PRDX1 function. Intriguingly, we observed a prominent formation of cytosolic PRDX1 condensates in both HRECs and HEK293T cells depleted with *KIF11* (Fig. 4r, s and Supplementary Fig. 10a, b). PRDX1 has recently been reported to undergo LLPS, which has an inhibitory effect on its peroxidase activity[35]. We therefore asked whether loss of KIF11 triggers the LLPS of PRDX1. The fluorescence recovery after photobleaching (FRAP) analysis revealed that, in the presence or absence of KIF11, the PRDX1-EYFP puncta in HEK293T cells recovered comparably to 30% of the initial level within 160 s after photobleaching (Fig. 4t, u). Notably, however, the puncta size was larger following *KIF11* knockdown (Fig. 4v). Additionally, treatment of cells with 10% 1,6-hexanediol (1,6-HD), a reagent used to disrupt LLPS condensates[36], successfully dissolved the PRDX1-EYFP puncta regardless of KIF11 status (Fig. 4w).

According to previous reports, PRDX1 monomers undergo oxidation to form peroxidase-inactive homodimers, a transformation that facilitates assembly into chaperone-active polymers and triggers PRDX1 LLPS[37–40]. Consistently, non-reducing SDS PAGE revealed a shift of PRDX1 from low-molecular-weight (LMW) monomers to high-molecular-weight (HMW) dimers upon depletion of *KIF11* in HRECs (Fig. 4x, y). Taken together, these data identify PRDX1 as a core binding partner of KIF11, and disruption of their interaction promotes PRDX1 LLPS.

### KIF11 inhibits Src-mediated PRDX1 phosphorylation-driven LLPS via competitive binding

Given that the phosphorylation of PRDX1 at Tyr194 leads to its inactivation and the accumulation of $H_2O_2$[41], we next evaluated whether KIF11 influences PRDX1 phosphorylation. As expected, *KIF11* knockdown significantly increased the phosphorylation of PRDX1 at Tyr194 in HRECs (Fig. 5a, b). Since phosphorylation is well known to regulate protein LLPS[40], we asked whether PRDX1 phosphorylation promotes its LLPS in the absence of KIF11. The overexpressed non-phosphorylatable PRDX1 mutant, Tyr194Gln (Y194Q), showed reduced condensates and the shift of most HWM dimers to LMW monomers compared to wild-type protein in HRECs and HEK293T cells (Fig. 5c–h and Supplementary Fig. 10c–f).

Bioinformatic analysis predicted intrinsically disordered regions (IDRs), known drivers of LLPS[42], in the N- and C-terminal regions of PRDX1 (Fig. 5i). Unexpectedly, Y194Q mutation led to a slight upregulation, rather than downregulation, of the IDR confidence in the C-terminus of PRDX1 compared to the wild-type protein (Fig. 5i). Thus, substitution of Tyr194 with Gln did not lead to disruption of disordered regions, suggesting that phosphorylation might be a critical trigger for PRDX1 LLPS.

To further investigate the role of phosphorylation in PRDX1 LLPS, we treated HRECs with gradient concentrations of larotinib, a tyrosine kinase inhibitor[43]. We identified 100 μM as the minimum effective dose that robustly inhibited PRDX1 Tyr194 phosphorylation (Supplementary Fig. 10g, h). Treatment of *KIF11* knockdown HRECs with larotinib markedly reduced the levels of HMW PRDX1 dimers while increasing the levels of LMW PRDX1 monomers (Fig. 5j, k). Additionally, Co-IP assays revealed an increase in the interaction between Y194Q-mutant PRDX1 and KIF11 (Fig. 5l, m). These results demonstrate that KIF11 inactivation prevents the inhibition of PRDX1 phosphorylation, thereby impairing the peroxidase activity of PRDX1 via the induction of HWM-mediated LLPS.

A previous report indicated that Src, a tyrosine kinase, serves as a critical upstream regulator of PRDX1-Y194 phosphorylation[44], therefore, we asked whether Src participates in PRDX1 phosphorylation in HRECs depleted with *KIF11*. This was confirmed by the dose-dependent reduction of PRDX1 phosphorylation upon treatment with PP1 (Supplementary Fig. 10i, j), a Src kinase inhibitor[44]. We then treated HRECs with 100 μM PP1 (Supplementary Fig. 10i, j), which effectively diminished PRDX1 Tyr194 phosphorylation, leading to a shift from HWM dimers to LMW monomers, in KIF11-deficient HRECs (Fig. 5n, o).

We next performed miniTurboID-based proximity labeling to capture transient kinase-substrate (Src-PRDX1) interactions and test whether KIF11 modulates PRDX1 phosphorylation through regulating Src-PRDX1 interaction. We co-expressed Src-miniTurboID-FLAG and PRDX1-HA in HEK293T cells in the presence or absence of KIF11-FLAG (Fig. 5p). Biotinylated proteins were enriched from the lysates using streptavidin beads (Fig. 5p). Src-miniTurboID-FLAG successfully labeled PRDX1-HA with biotin compared to Vector-miniTurboID-FLAG

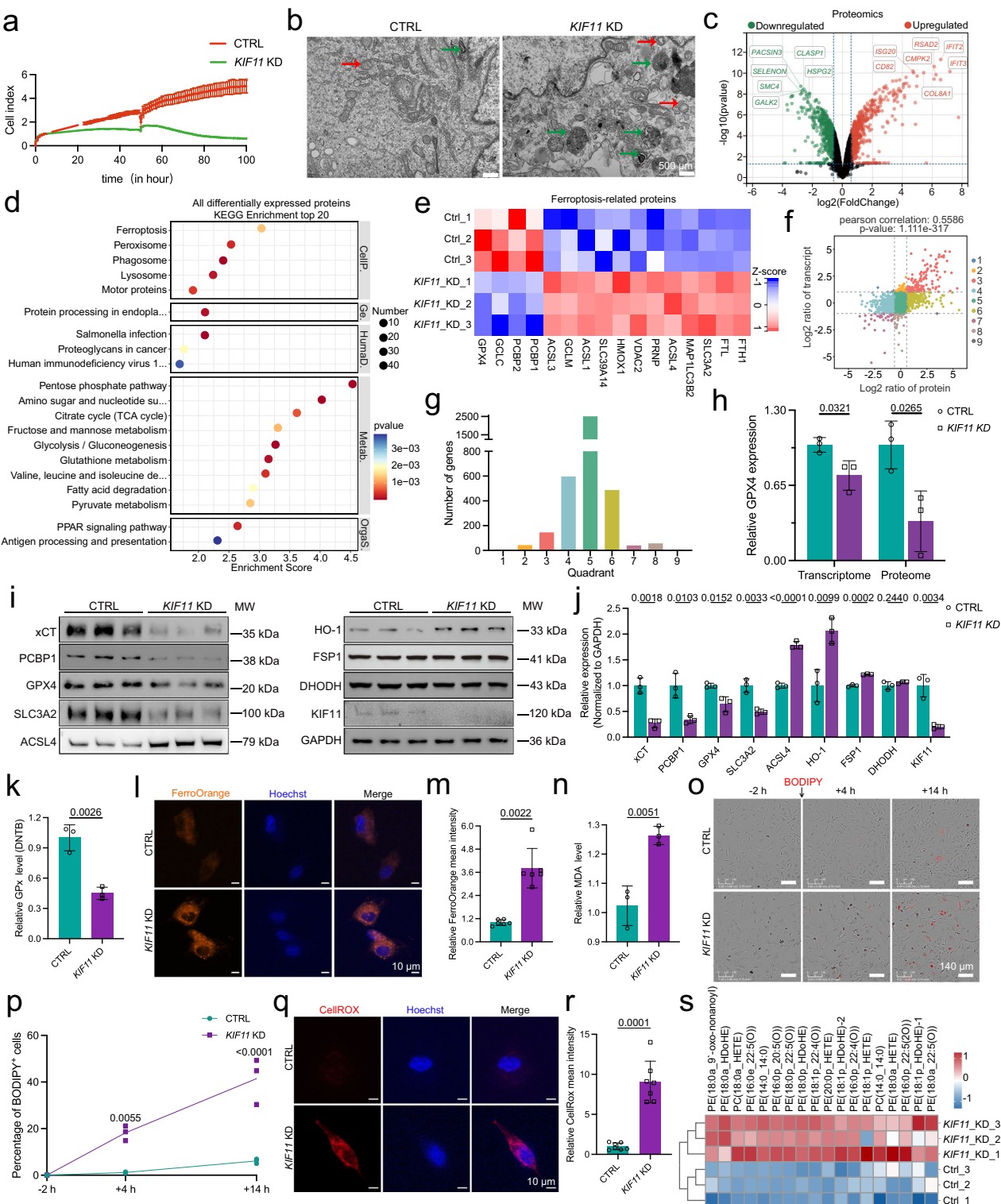

(Fig. 5q, r), confirming the interaction between Src and PRDX1. Interestingly, exogenous overexpression of KIF11 markedly decreased the amount of biotin-labeled PRDX1, along with reduced P-PRDX1 in cell lysates (Fig. 5s–u), suggesting that KIF11 competes with Src for PRDX1 binding and phosphorylation.

**Depletion of *PRDX1* or *CTNNB1* drives autophagy-accompanied ferroptosis that could be rescued by Fer-1**

Given that loss of KIF11 leads to impaired PRDX1 function, we tested whether knockdown of *PRDX1* phenocopies the defects associated

with KIF11 deficiency. Similarly, efficient *PRDX1* knockdown in HRECs exhibited decreased proliferation and subsequent death, accompanied by autophagy- and ferroptosis-like mitochondrial changes (Fig. 6a–g). *PRDX1* knockdown led to a ~40% reduction in GPx activity, with increased labile ferrous iron, oxidative stress, and lipid peroxidation (Fig. 6h–m), indicating that PRDX1 deficiency triggers autophagy-accompanied ferroptosis in HRECs.

We next performed bulk RNA-seq to analyze transcriptomic alterations following *PRDX1* knockdown (Fig. 6n and Supplementary Data 1). 3D PCoA revealed that *PRDX1*-knockdown and *KIF11*-

**Fig. 3 | KIF11-deficient HRECs exhibit signatures with increased ferroptosis.**
**a** Representative cell index values of CTRL and *KIF11*-depleted HRECs. *n* = 3.
**b** Representative transmission electron microscopy (TEM) images of CTRL and
*KIF11*-depleted HRECs. Green arrows, autophagosome; red arrows, mitochondria.
Scale bars, 500 nm. **c** Volcano plot of differentially expressed proteins in CTRL and
*KIF11* KD HRECs. **d** Bubble plot of the KEGG pathway enrichment analysis for dif-
ferentially expressed proteins upon *KIF11* knockdown. **e** Heatmap of the altered
ferroptosis-related genes in CTRL and *KIF11*-depleted HRECs. **f** Nine-quadrant dia-
gram integrating transcriptomic and proteomic data of CTRL and *KIF11*-depleted
HRECs, with log$_2$FoldChange thresholds of −1 and 1 for transcriptome, and −0.585
and 0.585 for proteome. **g** Number of genes in each quadrant. **h** Relative quanti-
fication of GPX4 mRNA and protein level in control and *KIF11*-depleted HRECs
profiled by transcriptome and Proteome. *n* = 3. Western blot (**i**) and relative
quantification (**j**) of ferroptosis-related proteins in control and *KIF11*-depleted
HRECs. *n* = 3. **k** Quantification of relative GPx activity in control and *KIF11*-depleted

HRECs. *n* = 3. Representative images (**l**) and relative mean intensity quantification
(**m**) of FerroOrange in CTRL and *KIF11*-depleted HRECs. Orange, FerroOrange; blue,
Hoechst. Scale bars, 10 µm. *n* = 6. **n** Relative MDA levels in CTRL and *KIF11*-depleted
HRECs. *n* = 3. Representative time-lapse images of BODIPY (red) (**o**) and quantifi-
cation of BODIPY$^+$ cells (**p**) in CTRL and *KIF11*-depleted HRECs. Scale bars, 70 µm.
*n* = 3. **q** Representative images of CTRL and *KIF11*-depleted HRECs stained with
CellROX and Hoechst. Red, CellROX; blue, Hoechst. Scale bars, 10 µm.
**r** Quantification of relative mean CellROX fluorescence intensity in CTRL and *KIF11*-
depleted HRECs. *n* = 7. **s** Heatmap of phospholipid peroxidation in CTRL and *KIF11*-
depleted HRECs. Data are presented as mean ± SD. *n* represents independent bio-
logical replicates. Statistical significance was determined using two-tailed Student's
*t* test (**h, j, k, n**), two-tailed Welch's t-test (**r**), Mann–Whitney U test (**m**), or two-way
ANOVA with Sidak's multiple comparisons test (**p**). Source data are provided as a
Source Data file.

knowledge cells clustered closely, indicating similar transcriptional
changes (Fig. 6o). Furthermore, the transcriptional profiles of *PRDX1*-
knockdown and *KIF11*-knockdown HRECs resembled those depleted
with *CTNNB1*, *FZD4*, *TSPAN12*, and *LRP5* (Fig. 6p). GSEA revealed a
strong negative correlation of *PRDX1*-knockdown cells with gene
clusters associated with the cell cycle and DNA replication, which were
validated at the mRNA and protein levels (Supplementary Fig. 11a–g).
Conversely, *PRDX1* knockdown positively correlated with autophagy,
ferroptosis, and lysosome signaling, with corresponding modulators
of autophagy and lysosome altered at both the protein and the mRNA
levels (Supplementary Fig. 12a–g), whereas ferroptosis modulators
were predominantly regulated at the protein level rather than the
mRNA level (Fig. 6q–s). The alterations of core ferroptosis-related
genes were validated by another shRNA (Supplementary Fig. 12h–j).
Moreover, immunofluorescence analysis revealed an increased num-
ber of LC3B puncta and elevated LAMP1 expression in PRDX1-deficient
cells (Supplementary Fig. 12k–m).

Consistent with those observed upon *KIF11* depletion, Fer-1
reversed *PRDX1* depletion-induced ferroptotic phenotypes, including
reduced viability, lipid peroxide accumulation, elevated ROS and labile
iron, decreased GPx activity, and dysregulated ferroptosis-related
proteins (Fig. 6t–z). These findings suggest that, similar to compro-
mised KIF11 function, PRDX1 deficiency drives autophagy-
accompanied ferroptosis in endothelial cells.

We next examined whether ferroptotic alterations related to KIF11
and PRDX1 deficiency could be recapitulated in CTNNB1-deficient
HRECs. *CTNNB1* knockdown induced cell death, lipid peroxidation,
accompanied by ferroptosis-like mitochondrial changes and altera-
tions in ferroptosis-associated proteins (Fig. 7a–h). Moreover, *CTNNB1*
depletion promoted formation of PRDX1 dimers, increased phos-
phorylation of PRDX1 at Tyr194 and cytosolic RPDX1 puncta, mirroring
the effects observed upon *KIF11* knockdown (Fig. 7i–n). Fer-1 treatment
significantly restored the cell death, lipid peroxidation, and changes in
ferroptosis-related proteins (Fig. 7o–u). Collectively, these findings
underscore a pathway in which impaired Norrin/β-catenin signaling
drives PRDX1 dysfunction-induced ferroptosis through KIF11.

## Inhibition of KIF11 compromises mouse retinal vascular development

To determine the mechanisms of *KIF11*-associated retinal vascular
defects in vivo, we constructed an inducible endothelial cell-specific
*Kif11* knockout mouse model[19]. *Kif11*$^{flox/flox}$; *Pdgfb*-iCreER (*Kif11* cKO)
mice exhibited comparable body weight with control littermates at P7
(Supplementary Fig. 13a). In line with the findings of Wang et al.[9], *Kif11*
cKO mice presented compromised vascular progression and
decreased vascular density at P7 (Fig. 8a–c). However, leakage of nei-
ther erythrocytes nor the low-molecular-weight tracer Sulfo-N-
hydroxysuccinimide (NHS)-biotin was detectable in *Kif11* cKO mice

(Supplementary Fig. 13b, c). At P10, capillaries in the outer plexiform
layer (OPL) were fully developed in WT mice but largely absent in *Kif11*
cKO mice (Fig. 8d). In agreement with observations in HRECs, TEM
analysis of retinal ECs revealed increased autophagosome formation
and ferroptosis-like mitochondrial alterations upon *Kif11* knockout
(Fig. 8e), supporting the occurrence of autophagy-accompanied
ferroptosis.

Given the high abundance of vascular endothelial cells in mouse
lung tissue[45], we then investigated the molecular changes in the lung
lysates. *Kif11*-depletion in ECs led to significantly increased PRDX1
phosphorylation at Tyr194 (Fig. 8f, g). Notably, *Kif11* cKO mice pre-
sented alterations in protein expression suggestive of increased
autophagic flux (Fig. 8h, i) and ferroptosis (Fig. 8j, k). These findings
suggest that dysfunction of KIF11 in mouse ECs induces autophagy-
accompanied ferroptosis, possibly impairing defective retinal
vascularization.

## Ferroptosis inhibition or PRDX1 restoration partially rescues defects in mice

Given the rescue effect of Fer-1 on the dysfunction of β-catenin/KIF11/
PRDX1 axis in HRECs, we next examined whether Fer-1 treatment could
restore vascular defects in FEVR mice. As anticipated, daily adminis-
tration of Fer-1 from P2 to P6 partially rescued the delayed retinal
vascular development in *Kif11* cKO and *Ctnnb1* cKO mice at P7, without
affecting the body weight (Fig. 9a–g and Supplementary Fig. 14a, b).

Since KIF11 dysfunction inhibits the peroxidase activity of PRDX1
by triggering its phosphorylation at Tyr194, we asked whether
restoration of PRDX1 could reverse the impairment of endothelial
function. RTCA revealed that overexpression of either wild-type PRDX1
or the Y194Q mutant increased cell viability, with the Y194Q mutant
yielding a significant higher cell index than the wild-type, indicative of
enhanced peroxidase activity (Fig. 9h). Consistently, retro-orbital
venous sinus injection of a lentivirus carrying the Y194Q mutant sig-
nificantly improved vascular development in *Kif11* cKO mice compared
with vehicle treatment (Fig. 9i–l). In contrast, a lentivirus carrying wild-
type PRDX1 produced a minimal rescue effect (Fig. 9i–l). Neither
treatment affected the body weight (Supplementary Fig. 14c). These
findings further support the role of β-catenin/KIF11/PRDX1 deficiency-
induced ferroptosis in compromised retinal vascularization.

In summary, we propose that KIF11 serves as a core downstream
target of Norrin/β-catenin signaling. Loss-of-function mutations in
either *KIF11* or upstream components of this pathway impair the per-
oxidase activity of PRDX1 by releasing the inhibitory effect of KIF11 on
Src-mediated Tyr194-PRDX1 phosphorylation, thereby triggering
PRDX1 LLPS. This process induces autophagy-accompanied ferropto-
sis in retinal endothelial cells, ultimately causing defective retinal
vascular development and contributing to the pathogenesis of
FEVR (Fig. 9m).

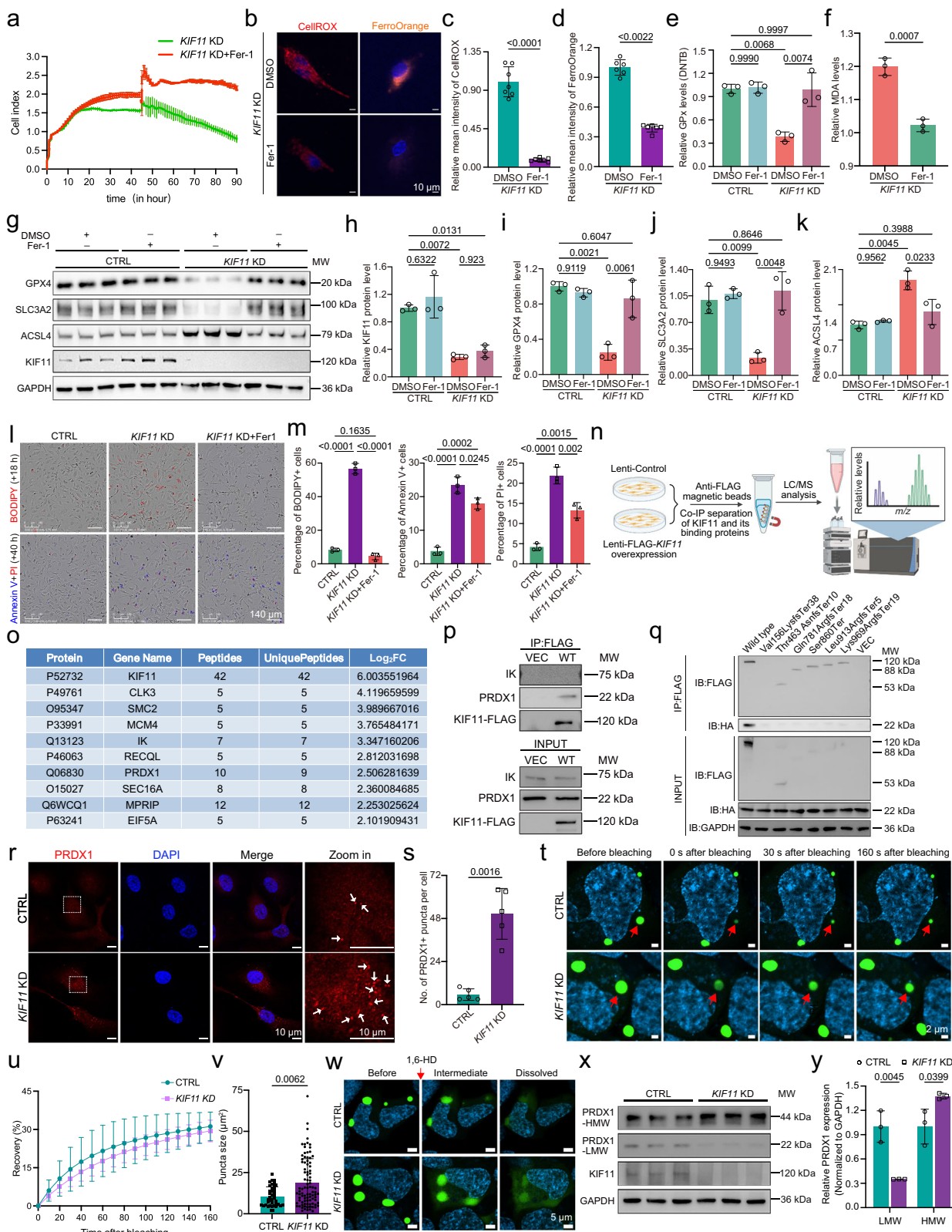

## Discussion

Mutations in the core upstream elements of the endothelial Norrin/β-catenin signaling pathway account for approximately 40% of FEVR cases[1,46]. However, owing to the limited knowledge of downstream regulators of β-catenin signaling in retinal endothelial cells, the development of effective therapeutic approaches for FEVR remains challenging. In this study, we identified KIF11 as a transcriptional downstream target of the Norrin/β-catenin signaling pathway. Considering that *KIF11* mutations have been implicated in FEVR phenotypes[47], that endothelial stabilization of β-catenin in *Kif11* knockout mice fails to restore retinal vascularization[9], and the observations in this study, it is plausible to conclude that KIF11 is a key downstream target of the Norrin/β-catenin pathway.

**Fig. 4 | Protective effect of Fer−1 and the role of KIF11–PRDX1 interaction in PRDX1 LLPS. a** Representative cell index values of *KIF11*-depleted HRECs treated with Fer-1 or DMSO. *n* = 3. Representative immunofluorescence images (**b**) and quantification of CellROX (**c**) and FerroOrange (**d**) intensities in *KIF11*-depleted HRECs treated with Fer-1 or DMSO. Red, CellROX; orange, FerroOrange; blue, Hoechst. Scale bars, 10 μm. *n* = 7 (**c**) and 6 (**d**). GPx activity (**e**) and MDA levels (**f**) in HRECs treated with Fer-1 or DMSO. *n* = 3. Immunoblotting (**g**) and quantification (**h**–**k**) of core ferroptosis-related proteins in HRECs treated with Fer-1 or DMSO. *n* = 3. Representative images (**l**) and quantification (**m**) of BODIPY (red), Annexin V (blue), and PI (red) staining in HRECs treated with DMSO or Fer-1. *n* = 3. **n** Schematic diagram of immunoprecipitation-mass spectrometry analysis. Created in BioRender. Li, S. (2026) https://BioRender.com/b75j366. **o** Top nine potential KIF11-binding partners identified by mass spectrometry. Co-IP analysis of interactions between KIF11-FLAG and IK or PRDX1 (**p**), and between mutant KIF11-FLAG and PRDX1-HA (**q**). **r** Representative images of the PRDX1 puncta in CTRL and *KIF11*-depleted HRECs. Dotted boxes: zoomed-in areas. Arrows, PRDX1 puncta. Red,

PRDX1; blue, DAPI. Scale bars, 10 μm. **s** Quantification of PRDX1 puncta numbers per cell. *n* = 5. Representative time-lapse FRAP images (**t**), mean percentage recovery of PRDX1-EYFP puncta (**u**), and quantification of PRDX1-EYFP puncta sizes (**v**) in CTRL and *KIF11*-depleted HEK293T cells transfected with PRDX1-EYFP. Arrows, the bleached puncta. Green, EYPF; blue, Hoechst. Scale bars, 2 μm. *n* = 3 (**u**). *n* = 39 (CTRL) and *n* = 91 (*KIF11* KD) cells pooled from three independent biological preparations (**v**). The error bar center represents the mean (**v**). **w** Representative images of PRDX1-EYFP overexpressing CTRL and *KIF11*-depleted HEK293T cells before and after 10% 1,6-HD treatment. Green, EYFP; blue, Hoechst. Scale bars, 5 μm. Non-reducing SDS PAGE immunoblot (**x**) and quantification (**y**) of the HWM- and LWM- PRDX1. *n* = 3. Data are presented as mean ± SD. Unless otherwise specified, *n* represents independent biological replicates. Statistical significance was determined using two-tailed Student's *t* test (**y**), two-tailed Welch's *t*-test (**c**, **f**, **s**), Mann−Whitney U test (**d**, **v**), one-way ANOVA with Tukey's multiple comparisons test (**m**), or two-way ANOVA with Tukey's multiple comparisons test (**e**, **h**–**k**). Source data are provided as a Source Data file.

KIF11 is well known for its role as a motor protein involved in spindle assembly and elongation during mitosis[48]. Here, we demonstrate that, in addition to regulating mitosis, KIF11 interacts with PRDX1 and preserves its antioxidant activity to prevent autophagy-accompanied ferroptosis. PRDX1 functions as a critical antioxidant that mitigates lipid peroxidation by regulating cellular iron and ROS levels[49,50]. Several studies have highlighted the role of PRDXs in protecting endothelial cells from ROS-induced cytotoxicity[51]. Overexpression of *prdx1* promotes vascular development in zebrafish, whereas its loss decreases sprouting from the caudal vein[52]. Together, these studies highlight a critical role for the PRDX family in vascular development and homeostasis. Consistently, we show that loss of PRDX1 in HRECs leads to autophagy-accompanied ferroptosis and activates a transcriptional program closely resembling that of cells with knockdown of *KIF11* or *CTNNB1*. These findings underscore a pivotal cascade connecting impaired Norrin/β-catenin signaling to PRDX1 dysfunction via KIF11.

The phosphorylation of PRDX1 has been reported to negatively affect its peroxidase activity[53]. We demonstrate that KIF11 binding to PRDX1 blocks Src-mediated PRDX1 phosphorylation at Tyr194, thereby maintaining PRDX1 antioxidant activity against oxidative stress. Notably, we also observed that the loss of KIF11 triggers the multimerization and cytosolic LLPS of PRDX1, a process which was recently reported to inhibit its peroxidase activity[54]. Although the link between PRDX1 phosphorylation and LLPS was previously unclear, our mutagenesis experiments, combined with a phosphorylation inhibitor, demonstrate that PRDX1 LLPS is triggered by Src-mediated phosphorylation.

Interestingly, bioinformatic predictions revealed a slightly higher IDR score for the phosphorylation-resistant PRDX1 mutant Y194Q compared with wild-type PRDX1, an unexpected result since higher IDR scores are typically linked to increased LLPS[42]. This raises the question of how phosphorylation regulates PRDX1 LLPS. Phosphorylation has been proposed to enhance electrostatic interactions within IDRs, facilitating self-association and driving LLPS[40,55]. Notably, phosphorylation-driven LLPS have been observed in other diseases, including TDP-43 in amyotrophic lateral sclerosis[56], Tau in Alzheimer's disease[55], and HIP-55 in heart failure[40]. However, currently available tools are unable to predict the IDRs of phosphorylated proteins. Thus, further investigations are needed to determine whether the phosphorylation of PRDX1 at Tyr194 contributes to enhanced electrostatic interactions.

The retina is particularly susceptible to oxidative damage due to its high oxygen consumption, abundant oxidizable components, and limited antioxidant defenses[57]. The relationship between excessive ROS and ferroptosis has been investigated in depth for decades and is involved in the pathogenesis of retinal vascular disorders, including age-related macular degeneration[58], proliferative diabetic

retinopathy[59], and retinopathy of prematurity[60]. Here, we demonstrate the involvement of oxidative stress and ferroptosis in the pathogenesis of FEVR. Due to limited patient samples, ferroptosis and PRDX1 LLPS could not be directly confirmed in human tissues. Nevertheless, Fer-1 treatment or overexpression of non-phosphorylatable PRDX1 partially improved endothelial functions in vivo and in vitro, suggesting potential therapeutic avenues for FEVR, though their efficacy in patients remains to be determined.

A recent work indicated that siRNA treatment can induce significant sensitization to ferroptosis independent of target gene knockdown[61]. Accordingly, a limitation of our work is that we were unable to generate stable *KIF11* knockout cells using the CRISPR-Cas9 system, owing to the rapid proliferation arrest and increased cell death upon *KIF11* depletion. Nevertheless, in this study, a lentiviral vector expressing a non-targeting shRNA was used as a negative control, thereby minimizing the potential confounding effects of RNAi treatment itself on ferroptosis sensitization.

In summary, we identified KIF11 as a core transcriptional target of the Norrin/β-catenin signaling pathway and uncovered a previously unrecognized interaction between KIF11 and PRDX1 that prevents Src-mediated phosphorylation-induced PRDX1 LLPS. Additionally, by linking autophagy-accompanied ferroptosis to the pathogenesis of FEVR, these findings provide mechanistic insights into how impaired Norrin/β-catenin signaling disrupts vascular development and indicate that targeting ferroptosis or restoring PRDX1 function may represent a potential therapeutic strategy for FEVR and other vascular diseases.

## Methods
### Mice
The *Kif11*-flox mice (Strain S-CKO-03251, Cyagen) and *Ctnnb1*-flox mice (VSM4101958, Viewsolid) were crossed with *Pdgfb-iCre-ER* mice[62] to obtain vascular endothelial cell-specific *Kif11* or *Ctnnb1* knockout mice, namely *Kif11*<sup>loxp/loxp</sup>; *Pdgfb-iCre-ER* (*Kif11* cKO) and *Ctnnb1*<sup>loxp/loxp</sup>; *Pdgfb-iCre-ER* (*Ctnnb1* cKO). Mice were induced through daily intraperitoneal injection with 50 μg tamoxifen (T6906, Topscience) from postnatal day 1 for 3 consecutive days. *Tspan12* knockout (*Tspan12* KO, Strain S-KO-08729, Cyagen) mice were used for single-cell RNA sequencing analysis. All mice were on a C57BL/6 background. The mice were maintained in an SPF animal facility under a controlled environment with a 12-h light/dark cycle, free access to food and water, at Sichuan Provincial People's Hospital. The ambient temperature was maintained at 22–26 °C with a relative humidity of 40–70%. The primers used for genotyping are shown in Supplementary Table 2.

For rescue experiments, lentiviruses (20 μl at a titer of $1 \times 10^9$ TU/mL) expressing KIF11, wild-type PRDX1, mutant PRDX1, or an empty vector (control) were obtained from Genechem Co. (Shanghai, China). For retro-orbital venous sinus injection[63], the P1 mice were anesthetized with isoflurane using a mouthpiece. A 31 G needle was inserted

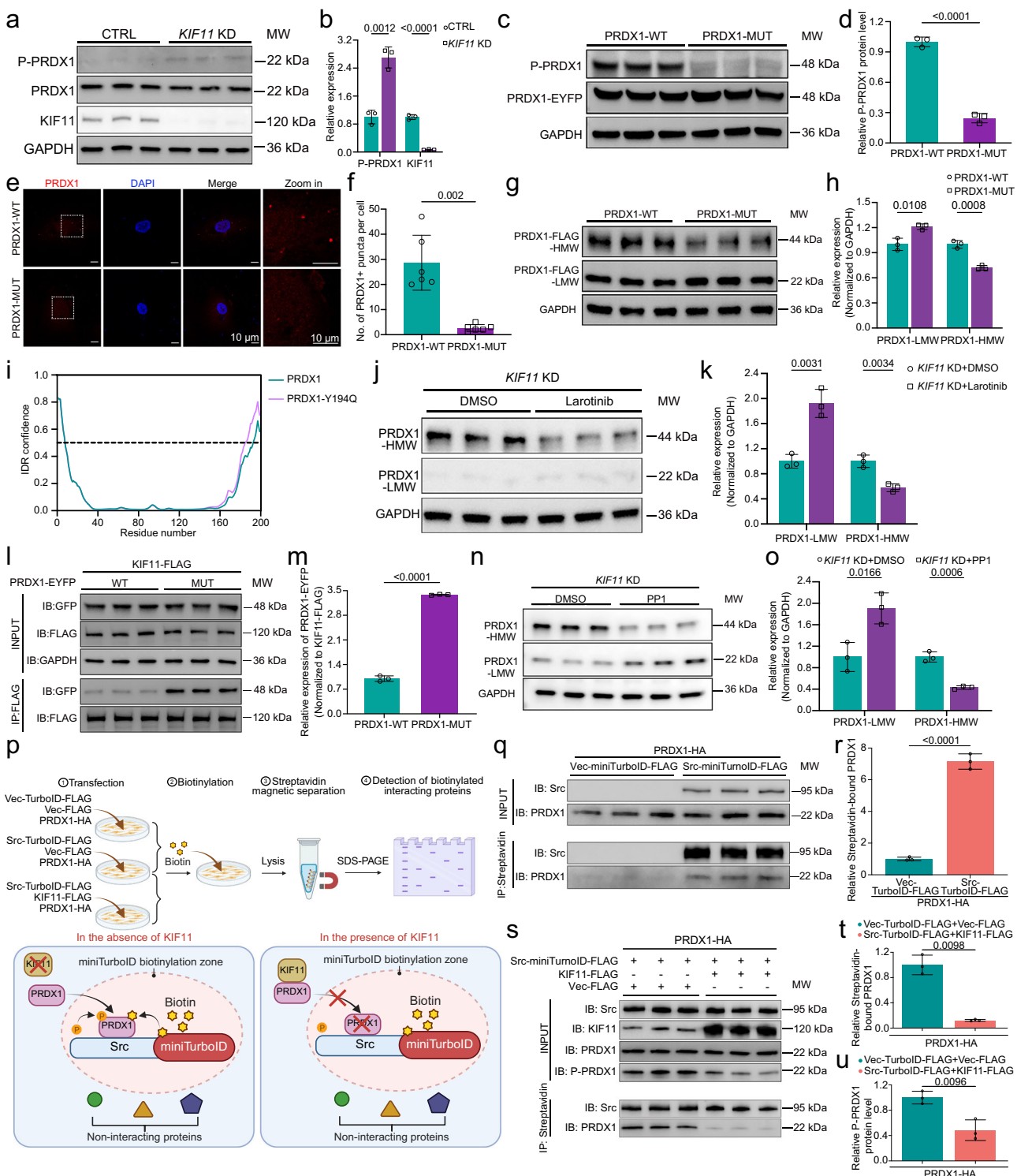

into the area of the retro-orbital sinus at an angle of approximately 40°. After a gentle injection, the needle was withdrawn slowly to avoid backflow. GFP fluorescence was used as an indicator of successful injection and efficient endothelial transduction. For rescue experiments targeting ferroptosis, mice were administered Fer-1 (30 mg/kg) or DMSO via intraperitoneal injection from P2 to P6. Both male and female mice were used for all experiments. Since familial exudative vitreoretinopathy is an inherited disorder driven by specific genetic mutations that do not exhibit documented sex-dependent phenotypic bias, sex was not considered as a primary biological variable in this study, and data from both sexes were pooled for analysis.

## Retinal vascular cell isolation and Single-cell RNA-sequencing

Due to the limited number of mice per litter, the wild-type samples were not derived from the same litters as the *Ctnnb1* cKO or *Tspan12* KO mice. Retinal tissues from five P6 mice per litter with the same genotype were immediately transferred to 5 mL centrifuge tubes and rinsed with 1640 medium containing 1% bovine serum albumin (BSA). The tissue was finely minced and enzymatically digested at 37 °C for 12 min using collagenase type II (4 mg/ml) and DNase I (0.05 mg/ml). The resulting suspension was filtered through a 70 μm cell strainer to remove debris and aggregated material. Cells were pelleted by centrifugation at 500 × *g* for 10 min, and single-cell suspensions were

**Fig. 5 | KIF11 inhibits Src-mediated PRDX1 phosphorylation-driven LLPS via competitive binding.** Western blot analysis (**a**) and quantification (**b**) of phosphorylated PRDX1-Tyr194 and KIF11 expression levels. $n = 3$. Western blot analysis (**c**) and quantification (**d**) of phosphorylated PRDX1-Tyr194 expression levels in HEK293T cells expressing WT or Y194Q-mutant PRDX1. $n = 3$. **e** Representative images of PRDX1 in WT or Y194Q-mutant PRDX1-overexpressed HRECs. Dotted boxes, zoomed-in regions. Red, PRDX1; blue, DAPI. Scale bars, 10 μm. **f** Quantification of PRDX1 puncta numbers per cell in HRECs overexpressing with WT or Y194Q-mutant PRDX1. $n = 6$. Non-reducing SDS PAGE immunoblot (**g**) and quantification (**h**) of HWM- and LWM- PRDX1 levels in WT and Y194Q-mutant PRDX1 overexpressed HRECs. $n = 3$. **i** Prediction of intrinsically disordered regions (IDRs) of WT and Y194Q-mutant PRDX1 using ESpritz (http://old.protein.bio.unipd.it/espritz/). Western blotting (**j**) and quantification (**k**) of HWM- and LWM-PRDX1 in *KIF11* KD HRECs treated with larotinib (100 μM) or DMSO. $n = 3$. Co-IP assays showing the interaction between KIF11-FLAG and either wild-type or Y194Q-mutant

PRDX1 (**l**), and relative quantification of PRDX1 binding (**m**). $n = 3$. Western blotting (**n**) and quantification (**o**) of HWM- and LWM-PRDX1 in *KIF11* KD HRECs treated with Src kinase inhibitor PP1 (100 μM) or DMSO. $n = 3$. **p** Schematic diagram of Co-IP assays of Src-miniTurboID-FLAG and PRDX1-HA in the presence or absence of KIF11. Created in BioRender. Li, S. (2026) https://BioRender.com/b75j366. Co-IP assays of Src-miniTurboID-FLAG and PRDX1-HA (**q**), and quantification of streptavidin-bound biotinlated-PRDX1-HA (**r**). $n = 3$. **s** Co-IP assays of streptavidin-bound biotinylated-PRDX1-HA by Src-miniTurboID-FLAG. Quantification of streptavidin-pulled PRDX1 (normalized to Src) (**t**), and levels of P-PRDX1 (normalized to PRDX1) in total lysates (INPUT) (**u**), in the presence or absence of overexpressed KIF11-FLAG. $n = 3$. Data are presented as mean ± SD. $n$ represents independent biological replicates. Statistical significance was determined using two-tailed Student's $t$ test (**b, d, h, k, m, o, r, u**) or two-tailed Welch's $t$-test (**f, t**). Source data are provided as a Source Data file.

confirmed under a microscope. Vascular cells, including endothelial cells (ECs) and microglia, were enriched using CD31 and CD11b MicroBeads (130-097-418 and 130-126-725, Miltenyi Biotec) following the manufacturer's protocol. Sorted cells were resuspended in culture medium for inspection. Single-cell libraries were prepared using the Chromium Next GEM Single Cell 3′ Reagent Kits v3.1 (10× Genomics) and sequenced on an Illumina NovaSeq 6000 platform (PE150) by OE Biotech (Shanghai, China).

Public scRNA-seq data of P6 WT, P26 WT and *Tspan12* KO retinas, as well as P60 WT and P60 *Ndp* KO retinas, were retrieved from the Gene Expression Omnibus (GEO) database (GSE175895, GSE213887, and GSE125708). FASTQ sequencing reads were aligned to the mouse reference genome (mm10-3.0.0) using Cell Ranger software (version 9.0.0, 10× Genomics), which also generated counts of unique molecular identifiers (UMIs) per cell barcode. The resulting UMI count matrices were processed in R using the Seurat package (version 4.0.0) for downstream analyses. Cells with fewer than 250 detected genes and 500 UMIs, or >15% of reads mapping to mitochondrial genes, were removed.

Gene expression normalization was performed with the NormalizeData function using the "LogNormalize" method, which scales the total expression per cell to 10,000, followed by log transformation. Highly variable genes (HVGs, $n = 2,000$) were identified using FindVariableGenes with FastExpMean and FastLogVMR as the mean and dispersion functions, respectively. Dimensionality reduction was conducted via PCA using the RunPCA function. To correct batch effects, Harmony integration was applied using RunHarmony from the Harmony R package (version 1.0). Graph-based clustering was performed with FindClusters, and a two-dimensional visualization of cellular states was generated using UMAP through RunUMAP. Cluster-specific marker genes were identified using FindAllMarkers with test.use = presto. Differential expression analysis between groups was carried out with FindMarkers using the same test, and $p$-values were adjusted with the Bonferroni method.

### Patient recruitment and clinical evaluation

The following characteristics were recorded as potential covariates to ensure a comprehensive understanding of the genotype-phenotype correlations: (1) Age: the cohort includes participants ranging from 4 months to 53 months. (2) Sex: the cohort includes 5 males and 1 female. While age and sex were collected for all participants, they were not treated as primary experimental variables, as familial exudative vitreoretinopathy is an inherited vascular disorder driven by genetic mutations with no documented sexual dimorphism or sex-dependent phenotypic bias in its clinical presentation. Patients were evaluated at the Xinhua Hospital, Shanghai Jiaotong University School of Medicine. Written informed consent was obtained from all participants or the legal guardians of minors. Participants did not receive financial compensation for their involvement in this study. The diagnostic criteria

for familial exudative vitreoretinopathy (FEVR) primarily consist of the following: (1) full-term birth with normal birth weight and no history of oxygen support; (2) presence of an avascular peripheral retina; (3) displaced or dragged retinal vessels and macula; (4) retinal folds (falciform) or retinal detachment; (5) neovascularization or subretinal exudates.

### Targeted gene sequencing and Sanger sequencing

Genomic DNA was isolated from peripheral blood using the QIAamp DNA Mini Kit (Qiagen), following the manufacturer's guidelines. DNA quantification was performed with a Nanodrop 2000 (Thermo Fisher Scientific). Approximately 1–3 μg of genomic DNA was then fragmented to an average fragment size of 150 bp using an S220 Focused-ultrasonicator (Covaris). Library preparation was carried out with the DNA Sample Prep Reagent Set (MyGenostics), involving end repair, adapter ligation, and PCR amplification. Amplified DNA was then enriched using the GenCap OT004/OT021A capture kit (MyGenostics). Biotin-labeled capture probes were designed in 100 bp that cover coding exons along with 50 bp flanking regions of all target genes. The capture process was executed as per the manufacturer's instructions. Finally, the enriched libraries were sequenced on an Illumina Nextseq 500 platform (Illumina).

The raw data were saved in FASTQ format following sequencing. Illumina sequencing adapters and low-quality reads (<80 bp) were removed using Cutadapt (http://code.google.com/p/cutadapt/). The filtered, clean reads were then aligned to the UCSC hg19 human reference genome with BWA. Duplicate reads were eliminated with Picard tools (http://broadinstitute.github.io/picard/), and the aligned reads were subsequently analyzed for variant detection. SNP and InDel variants were identified using HaplotypeCaller in GATK and filtered using VariantFiltration in GATK based on the following criteria: (a) mapping quality of variants <30; (b) total mapping quality zero reads <4; (c) approximate read depth <5; (d) QUAL score <50.0; (e) Fisher's exact test phred-scaled $p$-value for strand bias >10.0. Finally, the data were converted to VCF format. Variants were annotated using ANNOVAR and cross-referenced with multiple databases, including 1000 Genomes, ESP6500, dbSNP, EXAC, MyGenostics Inhouse, HGMD, and further analyzed using the prediction tool of MutationTaster. The candidate pathogenic variants were validated using Sanger sequencing and analyzed for cosegregation within families. PCR products were sequenced on an ABI 3730XL DNA Sequencer (Applied Biosystems, Thermo Fisher Scientific). The primers for validation were listed in the Supplementary Table 3.

### Cell culture, lentivirus-mediated knockdown, real-time cell analysis, and time-lapse live-cell imaging

Human microvascular endothelial cells (HRECs, ACBRI-181, Cell-systems) were cultured in a complete classic medium (4Z0-500, Cell-systems). HEK293T cells (CRL-3216™, ATCC) and HT1080 cells (CCL-

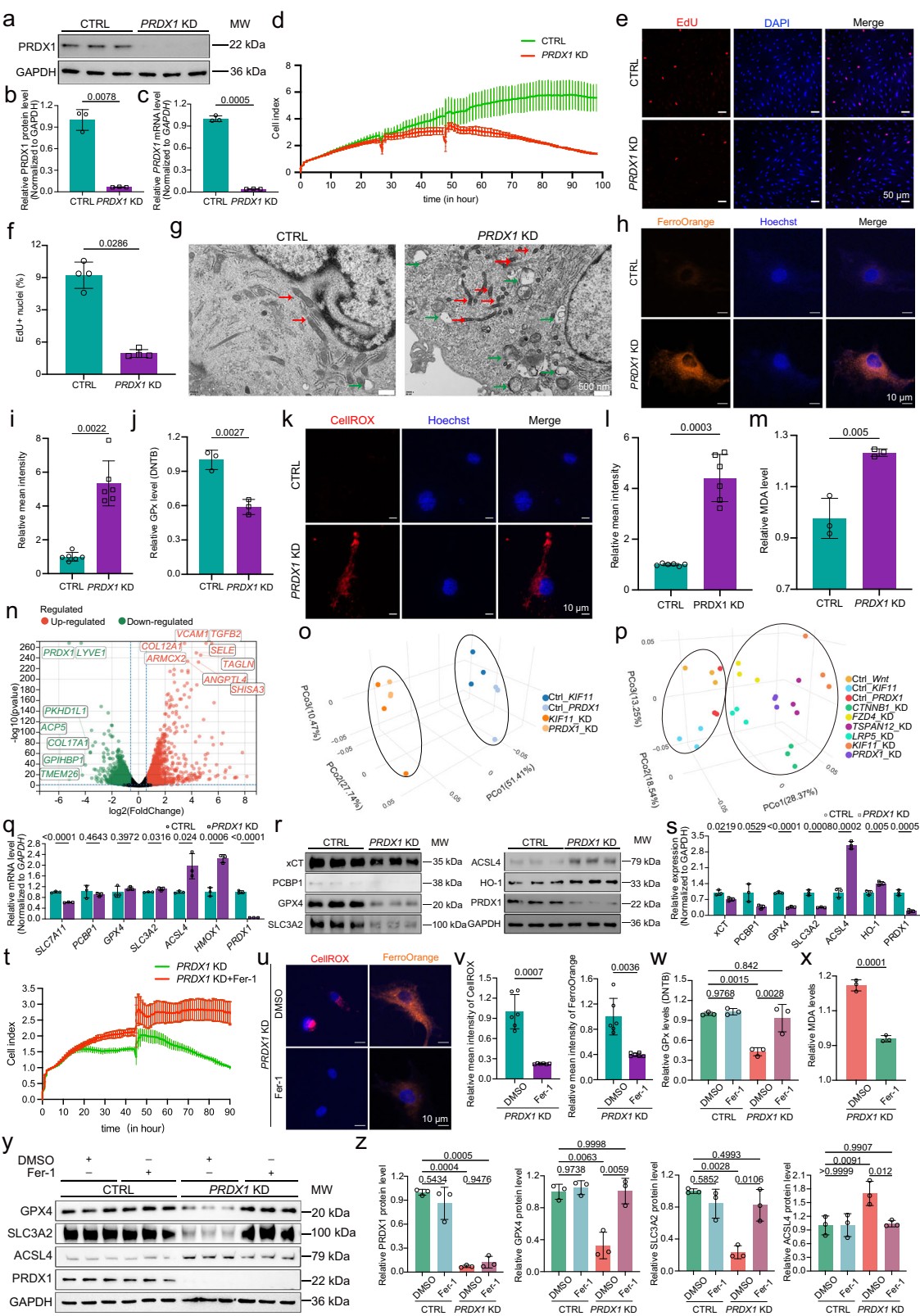

121™, ATCC) were cultured in Dulbecco's Modified Eagle Medium (DMEM) (11995040, Gibco) and Minimum Essential Medium (MEM), respectively. Both media were supplemented with 10% fetal bovine serum (C04001-500, VivaCell) and 1% penicillin/streptomycin (15070063, Thermo Fisher Scientific). Gene knockdown was performed using lentivirus-mediated shRNA obtained from Genechem Co. (Shanghai, China). The shRNA sequences are listed in

Supplementary Table 4. The Agilent xCELLigence Real-Time Cell Analysis systems[64] and xCELLigence RTCA eSight were used for live cell analysis and time-lapse live-cell imaging. Briefly, for time-lapse live-cell imaging, cells were seeded on E-plate VIEW 24 h after lentivirus infection. At 24 h post-seeding, 50 μl of culture medium was replaced with medium containing lipid peroxidation sensor BODIPY™ 665/676 (B3939, Thermo Fisher Scientific), Annexin V (A35122, Thermo Fisher

**Fig. 6 | Loss of PRDX1 in HRECs induces ferroptosis that could be restored by Fer-1.** Western blot (**a**) and quantification of PRDX1 protein (**b**) and mRNA levels (**c**) upon *PRDX1* KD. $n = 3$. Representative cell index values (**d**), EdU images (**e**), and quantification (**f**) of EdU$^+$ cells (%) in *PRDX1*-depleted HRECs. Red, EdU; blue, DAPI. Scale bars, 50 μm. $n = 4$. **g** TEM images in CTRL and *PRDX1*-depleted HRECs. Green arrows, autophagosome; red arrows, mitochondria. Scale bars, 500 nm. Representative FerroOrange images (**h**) and intensity quantification (**i**) in CTRL and *PRDX1*-depleted HRECs. Orange, FerroOrange; blue, Hoechst. Scale bars, 10 μm. $n = 6$. Relative GPx activity ($n = 3$) (**j**), CellROX immunofluorescence images (**k**) and mean intensity quantification ($n = 6$) (**l**), and MDA levels ($n = 3$) (**m**) in CTRL and *PRDX1* KD HRECs. Red, CellROX; blue, Hoechst. Scale bars, 10 μm. **n** Volcano plots of DEGs upon *PRDX1* KD. 3D PCoA plot summarizing bulk RNA-seq sample distribution of CTRL, *KIF11*, and *PRDX1* KD HRECs (**o**), and of CTRL and all knockdown groups (**p**). **q** Quantification of ferroptosis-related gene mRNAs in CTRL and *PRDX1*-depleted HRECs. $n = 3$. Western blot (**r**) and quantification (**s**) of ferroptosis-related proteins upon *PRDX1*-depleted. $n = 3$. **t** Representative cell index values of *PRDX1*-depleted HRECs treated with Fer-1 or DMSO. $n = 3$. Representative immunofluorescence (**u**) and quantification of CellROX (**v**, left) and FerroOrange (**v**, right) in *PRDX1*-depleted HRECs treated with Fer-1 or DMSO. Red, CellROX; orange, FerroOrange; blue, Hoechst. Scale bars, 10 μm. $n = 6$. GPx activity (**w**) and MDA levels (**x**) in *PRDX1*-depleted HRECs treated with Fer-1 or DMSO. $n = 3$. Immunoblotting (**y**) and quantification (**z**) of core ferroptosis-related proteins in CTRL and *PRDX1*-depleted HRECs treated with Fer-1 or DMSO. $n = 3$. Data are presented as mean ± SD. $n$ represents independent biological replicates. Statistical significance was determined using two-tailed Student's $t$ test (**j, m, q, s, x**), two-tailed Welch's t-test (**b, c, l, v**), Mann–Whitney U test (**f, i**), or two-way ANOVA with Tukey's multiple comparisons test (**w, z**). Source data are provided as a Source Data file.

Scientific), and Propidium Iodide (PI, HY-D0815, Med Chem Express). Time-laspe imaging was performed with scans acquired every 30 min. Four micrometers Fer-1 (HY-100579, Med Chem Express) or DMSO was added to the culture medium 24 h after lentivirus infection. Fifty nanograms/milliliter Norrin (3014-NR-025, R&D) was added to the culture medium 72 h after lentivirus infection for 24 h.

### EdU labeling, TUNEL fluorescent assays, ROS staining, intracellular iron measurement, and LDH release assays

For EdU labeling, HRECs were treated with 10 μM EdU for 3 h and then harvested for staining using Click-iT EdU Alexa Fluor-594 Imaging Kit (C10339, Thermo Fisher Scientific). The TUNEL Apoptosis Detection Kit (40308ES20, YEASEN) was used for TUNEL apoptosis detection in HRECs. To detect the cellular ROS, the HRECs were incubated with the CellROX® Deep Red Reagent (C10422, Life Technologies) at a final concentration of 5 μM for 30 min at 37 °C, followed by three PBS rinses before microscopy. To detect intracellular Fe$^{2+}$, the HRECs were washed three times with HBSS and incubated with FerroOrange (1 μM) (F374, DOJINDO) for 30 min at 37 °C before microscopy. LDH release assays were performed Cytotoxicity LDH Assay Kit (HY-K1090, MedChenExpress) according to the manufacturer's instructions. The cultured HRECs were fixed in 4% PFA for 15 min and rinsed with PBS 3 times before being blocked with 5% fetal bovine serum. The antibodies used for immunofluorescence staining were listed in Supplementary Table 5. The images were obtained using a Zeiss LSM900 confocal microscope, Zeiss Axio Observer 7, and processed with Adobe Illustrator.

### Measure of lipid peroxidation malondialdehyde (MDA) and activity of total glutathione peroxidase in HRECs

The levels of MDA were measured according to the manufacturer's instructions (S0131S, Beyotime). Briefly, about $3 \times 10^6$ HRECs were lysed, and the supernatants were mixed with the working solution. Then, the mixtures were heated at 100 °C for 40 min, and the absorbance was detected by SuperMax 3000AL Microplate reader (FLASH) at 532 nm after cooling down to room temperature. The activity of cellular glutathione peroxidase was measured using a kit with DTNB (S0055S, Beyotime). The cells were lysed, and the supernatant was collected for subsequent procedures; the absorbance was measured using a spectrophotometer at 412 nm.

### Plasmids, site-directed mutagenesis, and transfection

Plasmids encoding full-length wild-type *KIF11* (NM_004523.4) with a FLAG tag, *PRDX1* (NM_002574.4) with an HA or EYFP tag, and plasmids expressing FLAG-tagged miniTurboID, with or without an N-terminal Src fusion, were purchased from Youbio (China). Plasmids encoding the KIF1 or PRDX1 (Tyr194Gln) mutants were synthesized by YouBio (Changsha, China). HEK293T cells were transfected with plasmids using Lipofectamine 3000 (L3000015, Invitrogen), and cells were collected 48 h after transfection.

### Fluorescence recovery after photobleaching (FRAP) assay

FRAP assay was performed on HEK293T cells transfected with wild-type or mutant PRDX1-EYFP using a Zeiss LSM900 confocal microscope equipped with an incubation chamber (5% CO$_2$, 37 °C). Cell nuclei were stained with Hoechst (C1027, Beyotime) for at least 20 min before imaging. Fluorescence recovery was recorded every 5 s for 5 min and analyzed using Zen microscopy software. Images were processed and quantified by the FRAP profiler v2 plugin using Image J (version 2.16.0).

To dissolve the LLPS condensates, 1,6-Hexanediol (1,6-HD, 240117, Sigma) was dissolved in the DMEM culture medium (10% concentrations) and warmed at 37 °C incubator before adding to cells. The LSM900 confocal microscope (Zeiss) was used to visualize live-cell imaging for cells treated with 10% 1,6-HD.

### RNA extraction, quantitative real-time PCR, and bulk RNA-sequencing

The total RNA was extracted from HRECs or lung tissues using Trans-Zol (ET101-01-V2, Transgen Biotech) and then reverse transcribed using the EasyScript One-Step RT-PCR SuperMix (AT411-02, TransGen Biotech). The quantitative real-time PCR was conducted using Trans-Start Tip Green qPCR SuperMix (AQ142-21, TransGen Biotech) on the ABI 7500. The primers were listed in Supplementary Table 6. The bulk RNA-sequencing transcriptome was conducted by Biozeron (Shanghai). Briefly, the libraries were constructed using TruSeq RNA Sample Preparation Kit (Illumina) with 1 μg of total RNA. mRNA was isolated through polyA selection using oligo(dT) beads and then fragmented with fragmentation buffer. cDNA synthesis, end repair, A-base addition, and adapter ligation were completed according to protocol. Libraries were size-selected to obtain cDNA fragments between 200–300 bp on 2% Low Range Ultra Agarose and then PCR-amplified with Phusion DNA polymerase (New England Biolabs) for 15 cycles. After quantification by TBS380, paired-end libraries were sequenced using the Illumina NovaSeqXplus or NovaSeq 6000 system (150 bp × 2, Shanghai BIOZERON Co., Ltd).

The raw paired-end reads were trimmed and quality-controlled with Trimmomatic (version 0.39). The cleaned reads were then aligned to the reference genome using HISAT2 (version 2.1.0). Data quality was assessed with Qualimap (version 2.2.1), and gene counts were generated using HTSeq (version 0.11.1).

For the knockdown experiments targeting *FZD4*, *LRP5*, *TSPAN12*, and *CTNNB1*, the same batch of cells infected with a lentivirus carrying a non-targeting shRNA was used as a negative control. The *KIF11* and *PRDX1* knockdown data set was compared to its own corresponding non-targeting shRNA control.

Differential expression analysis to identify DEGs was conducted using the edgeR package (version 3.34.0) based on the FPKM (Fragments Per Kilobase of transcript per Million mapped reads). The transcripts with a |log$_2$FoldChange| > 1 and $p$-value < 0.05 were classified as differentially expressed.

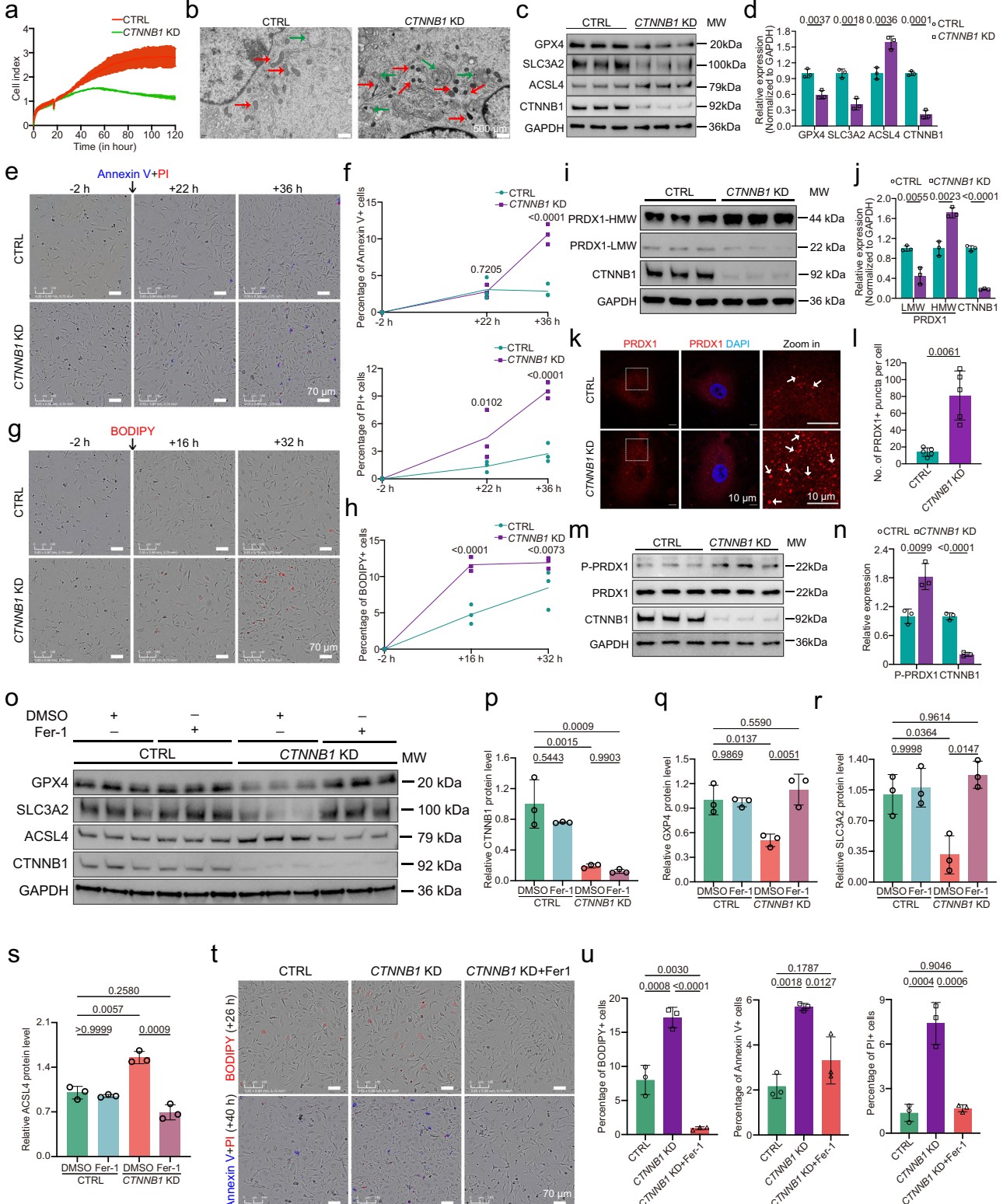

## Weighted gene co-expression network analysis (WGCNA), protein–protein interacting (PPI) network construction, and hub gene identification

The ComBat function in the sva R package (version 3.44.0) was used to combine different gene expression profiles by removing the batch effects based on FPKM[65]. WGCNA was conducted using the WGCNA R packages (version 1.71) to identify a correlation between modules and samples. Briefly, the Pearson correlation coefficient matrix was created from the integrated gene expression data and then was transformed into a weighted adjacency matrix. The topological overlap matrix (TOM) was created using the TOM similarity function and the TOM-based dissimilarity calculated by the "hclust" function. The hierarchical clustering dendrogram was then built based on TOM dissimilarity, and the modules were identified using a dynamic tree cut with a minimum module size of 30. Then, the module eigengene (ME) dissimilarity was calculated, and the cut-off for merging similar modules was set as 0.3.

**Fig. 7 | CTNNB1-deficient HRECs exhibit signatures with increased ferroptosis that could be restored by Fer-1. a** Representative cell index values of CTRL and *CTNNB1*-depleted HRECs measured using the xCELLigence RTCA system. *n* = 3. **b** Representative TEM images of CTRL and *CTNNB1*-depleted HRECs. Green arrows, autophagosome; red arrows, mitochondria. Scale bars, 500 nm. Western blot (**c**) and relative quantification (**d**) of ferroptosis-related proteins in CTRL and *CTNNB1*-depleted HRECs. *n* = 3. Representative time-lapse images (**e**) and quantification of percentages of Annexin V+ and PI+ cells (**f**). Scale bars, 70 μm. *n* = 3. Representative time-lapse images (**g**) and quantification of percentages of BODIPY+ cells (**h**). Scale bars, 70 μm. *n* = 3. **i** Non-reducing SDS PAGE immunoblot analysis of PRDX1 in CTRL and *CTNNB1*-depleted HRECs. **j** Quantification of the relative HWM- and LWM-PRDX1 and CTNNB1 expression levels in CTRL and *CTNNB1*-depleted HRECs. *n* = 3. **k** Representative images of the CTRL and *CTNNB1*-depleted HRECs stained with anti-PRDX1 and DAPI. Dotted boxes, zoomed-in regions. Arrows, PRDX1 puncta.

Red, PRDX1; blue, DAPI. Scale bars, 10 μm. **l** Quantification of PRDX1 puncta numbers per cell in CTRL and *CTNNB1*-depleted HRECs. *n* = 5. Western blot analysis (**m**) and quantification (**n**) of protein levels of phosphorylated PRDX1-Tyr194 and CTNNB1 in CTRL and *CTNNB1*-depleted HRECs. *n* = 3. Immunoblotting (**o**) and relative quantification (**p**–**s**) of core ferroptosis-related proteins in CTRL and *CTNNB1*-depleted HRECs treated with Fer-1 (4 μM) or DMSO. *n* = 3. Representative images (**t**) and quantification (**u**) of BODIPY (red), Annexin V (blue), and PI (red) staining in CTRL and *CTNNB1* KD HRECs treated with DMSO, and *CTNNB1*-depleted HRECs treated with Fer-1 (4 μM). *n* = 3. Data are presented as mean ± SD. *n* represents independent biological replicates. Statistical significance was determined using two-tailed Student's *t* test (**d, j, n**), two-tailed Student's Welch's t-test (**l**), one-way ANOVA with Tukey's multiple comparisons test (**u**), or two-way ANOVA with Sidak's multiple comparisons test (**f, h, p**–**s**). Source data are provided as a Source Data file.

The protein–protein interacting data for genes in the interested modules were extracted from the Search Tool for the Retrieval of Interacting Genes (STRING). Subsequently, CytoHubba-a plugin from Cytoscape (version 3.10.1) was utilized to identify hub genes using maximal clique centrality (MCC).

### Over-representation analysis (ORA)

Genes from the dark gray and blue modules of WGCNA, upregulated or downregulated genes from *KIF11* KD transcriptome, differentially expressed proteins from *KIF11* KD proteome, or differentially expressed proteins in quadrants 4 and 6 (|log₂FoldChange| < 0.585 for transcriptome, |log₂FoldChange| > 0.585 for proteome, $p < 0.05$) were enriched by KEGG pathways using the clusterProfiler (version 3.14.3).

### Gene set enrichment analysis (GSEA)

The GSEABase package (version 1.44.0) was used to load the gene set file, which was downloaded and processed from the KEGG database (https://www.kegg.jp/). Gene Set Enrichment Analysis (GSEA) was used to complete KEGG term enrichment analysis on genes detected in *KIF11* KD and *PRDX1* KD transcriptomes, and *KIF11* KD proteome.

### Nine-quadrant diagram for integrated transcriptome and proteome data

The log₂FoldChange and *p*-value of genes following *KIF11* knockdown in transcriptome and proteome data were integrated using a nine-quadrant diagram by the OmicShare tool (https://www.omicshare.com). The threshold for the division of the nine quadrants was |log₂FoldChange| > 1 for transcriptome and |log₂FoldChange| > 0.585 for proteome.

### ATAC-seq and ChIP-seq analysis

The raw data of ATAC-seq (GSE122117) and ChIP-seq (GSE164442) were obtained from the GEO database and analyzed as previously described[23]. Briefly, the reads were trimmed in silico to remove adapter sequences, low-quality reads, and 50-bp length using trimGalore (version 1.18) and aligned to the mm10 reference genome using the Bowtie2 alignment tool (version 2.3.5.1). Mitochondrial reads were filtered for the subsequent analyses. All peak calling of ATAC-seq data was performed using MACS2 (version 2.2.7.1) with callpeak-format BAMPE parameters. BigWig files were generated and normalized using RPKM by Deeptools (version 3.4.3). Heatmaps were created by Deeptools using the RPKM-normalized bigwig files.

### Western blot analysis and mass spectrometry analysis

Total proteins were extracted from HRECs or lung tissues using RIPA lysis buffer (R0010, Solarbio) and denatured using loading buffer (P1040, Solarbio). The proteins were separated by electrophoresis on 4–20% SDS PAGE and transferred onto the nitrocellulose membrane. The membranes were blocked with 5% skimmed milk in TBST for 1 h at room temperature and incubated with primary antibodies overnight at

4 °C, followed by incubation with secondary antibodies for 1 h at room temperature. The protein signals were detected using SuperSignal™ West Dura Extended Duration Substrate (34075, Thermo Fisher Scientific) by Touch Imager (e-BLOT) and analyzed using Image J software (version 2.16.0).

Mass spectrometry analysis was performed by Shanghai GENE-CHEM Co., Ltd and Applied Protein Technology. Briefly, total proteins for proteomic analysis were extracted from control and *KIF11*-depleted HRECs (*n* = 3 biological replicates per group). For IP-MS analysis, proteins were pulled down using anti-Flag magnetic beads from HRECs either overexpressing Flag-tagged KIF11 (*n* = 1) via lentiviral transduction or a non-overexpressing control (*n* = 1). All protein samples were sonicated and solubilized in SDT buffer (4% SDS, 100 mM Tris-HCl, pH 7.6). The samples were then boiled and centrifuged at 4 °C for 15 min to remove debris. The proteins were then reduced with 100 mM DTT for 5 min at 100 °C, and the UA buffer (8 M urea, 150 mM Tris-HCl, pH 8.5) was used to remove the detergent, DTT, and low-molecular-weight components. After blocking the reduced cysteine residues with 100 μl iodoacetamide (100 mM IAA in UA buffer), the protein solution was subsequently digested with 4 μg trypsin in 40 μl 50 mM NH₄HCO₃ buffer overnight at 37 °C. Peptides from each sample were desalted using C18 cartridges, concentrated by vacuum centrifugation, and reconstituted in 40 μL of 0.1% (v/v) formic acid. The peptide concentration was estimated by measuring the UV absorbance at 280 nm. An extinction coefficient of 1.1 for a 0.1% (w/v) solution was used, which was calculated based on the frequency of tryptophan and tyrosine residues in vertebrate proteins.

For proteomics, the LC-MS/MS analysis was conducted on an Orbitrap Exploris 480 or a Q Exactive mass spectrometer (Thermo Fisher Scientific) coupled to Easy nLC (Thermo Fisher Scientific). Two micrograms of peptides were loaded onto a C18 reversed-phase analytical column (Acclaim PepMap RSLC, 50 μm × 15 cm, nanoViper, P/N 164943; Thermo Fisher Scientific) in buffer A (0.1% formic acid) and separated with a linear gradient of buffer B (80% acetonitrile and 0.1% formic acid) at a flow rate of 300 nL/min. MS data were acquired using a data-dependent top-10 method, dynamically selecting the most abundant precursor ions from the survey scan (m/z 350–1200) for higher-energy collisional dissociation (HCD) fragmentation. Full MS1 scans were acquired at a resolution of 120,000 (at m/z 200) with an automatic gain control (AGC) target of 300% and a maximum injection time (IT) of 50 ms. The data-dependent mode was set to cycle time (1.5 s). MS2 scans were acquired at a resolution of 15,000 (at m/z 200) with an AGC target of 75% and a maximum IT of 35 ms. The isolation window was set to 1.6 m/z, and microscans were set to 1. Only ions with a charge state between 2 and 6 were selected for fragmentation. Dynamic exclusion was set to 30 s, and the normalized collision energy (NCE) was 33%.

For IP-MS, LC-MS/MS analysis was performed on a timsTOF Pro mass spectrometer (Bruker) coupled to an Evosep One liquid chromatography system (Denmark). The peptides were loaded onto a C18 reversed-phase analytical column (15 cm length, 150 μm inner

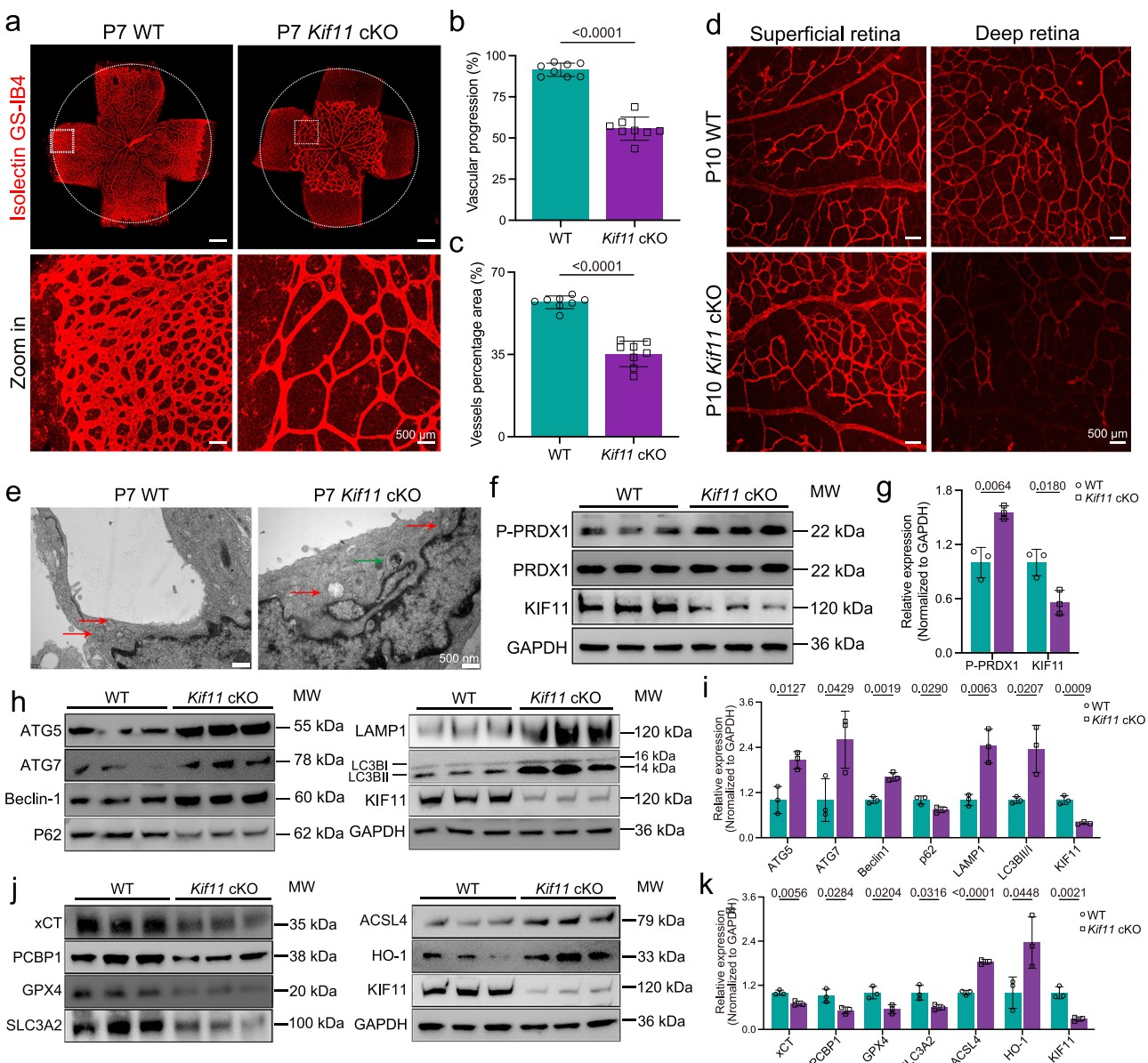

**Fig. 8 | Knockout of *Kif11* in mice vascular endothelial cells leads to retarded retinal vascular growth by promoting autophagy and ferroptosis signaling.** **a** Representative images of retinal flat mounts from P7 littermate control and *Kif11* cKO mice. The circles indicate vessel outgrowth of the control. Red, Isolectin GS-IB4. Dotted boxes indicate magnified areas. Scale bars, 500 μm and 50 μm. Quantification of retinal vascular progression (**b**) and vessel density (**c**) of P7 littermate control and *Kif11* cKO mice. $n = 8$. **d** Representative images of superficial and deep retinal vasculature from P10 littermate control and *Kif11* cKO mice. Red, Isolectin GS-IB4. Scale bars, 500 μm. **e** Representative TEM images of retinas from P7 littermate control and *Kif11* cKO mice. Green arrows, autophagosome; red arrows,

mitochondria. Scale bars, 500 nm. Western blot analysis (**f**) and quantification (**g**) of phosphorylated PRDX1-Tyr194 in lung lysates from P15 littermate control and *Kif11* cKO mice. $n = 3$. Immunoblotting analysis (**h**) and quantification of relative expression levels (**i**) of core autophagy-related genes in lung lysates from P15 littermate control and *Kif11* cKO mice. $n = 3$. Immunoblotting analysis (**j**) and quantification of relative expression levels (**k**) of core ferroptosis-related genes in lung lysates from P15 littermate control and *Kif11* cKO mice. $n = 3$, Data are presented as mean ± SD. $n$ represents independent biological replicates or the number of mice per group. Statistical significance was determined using a two-tailed Student's $t$ test (**b**, **c**, **g**, **i**, **k**). Source data are provided as a Source Data file.

diameter, 1.9 μm resin) in buffer A (0.1% formic acid in water) and separated with a linear gradient of buffer B (99.9% acetonitrile and 0.1% formic acid) at a flow rate of 220 nL/min. The mass spectrometer was operated in positive ion mode. The electrospray voltage applied was 1.6 kV. Precursors and fragments were analyzed using the TOF detector over a mass range of m/z 100–1700. The timsTOF Pro was operated in parallel accumulation serial fragmentation (PASEF) mode. PASEF mode data collection was performed based on the following parameters: the ion mobility coefficient ($1/K_0$) value was set from 0.75 to 1.35 V s/cm²; each cycle included 1 MS and 8 PASEF MS/MS scans. Active exclusion was enabled with a release time of 24 s.

The MS data were analyzed by MaxQuant (version 1.6.14) and searched against the Uniprot_homo_20221024_20402_9606_swiss_prot. The global false discovery rate (FDR) threshold for peptide and protein identification was 0.01. Protein abundance was quantified based on the normalized spectral protein intensity (LFQ intensity). Proteomics data contain a significant number of missing values that severely hinder proper differential analysis. Based on a previous work that suggested a proper imputation method[66], we first categorized the proteins into two groups: (1) one group includes proteins with at least two non-missing values within the same sample group; (2) the other group includes proteins with at least two non-missing values in one

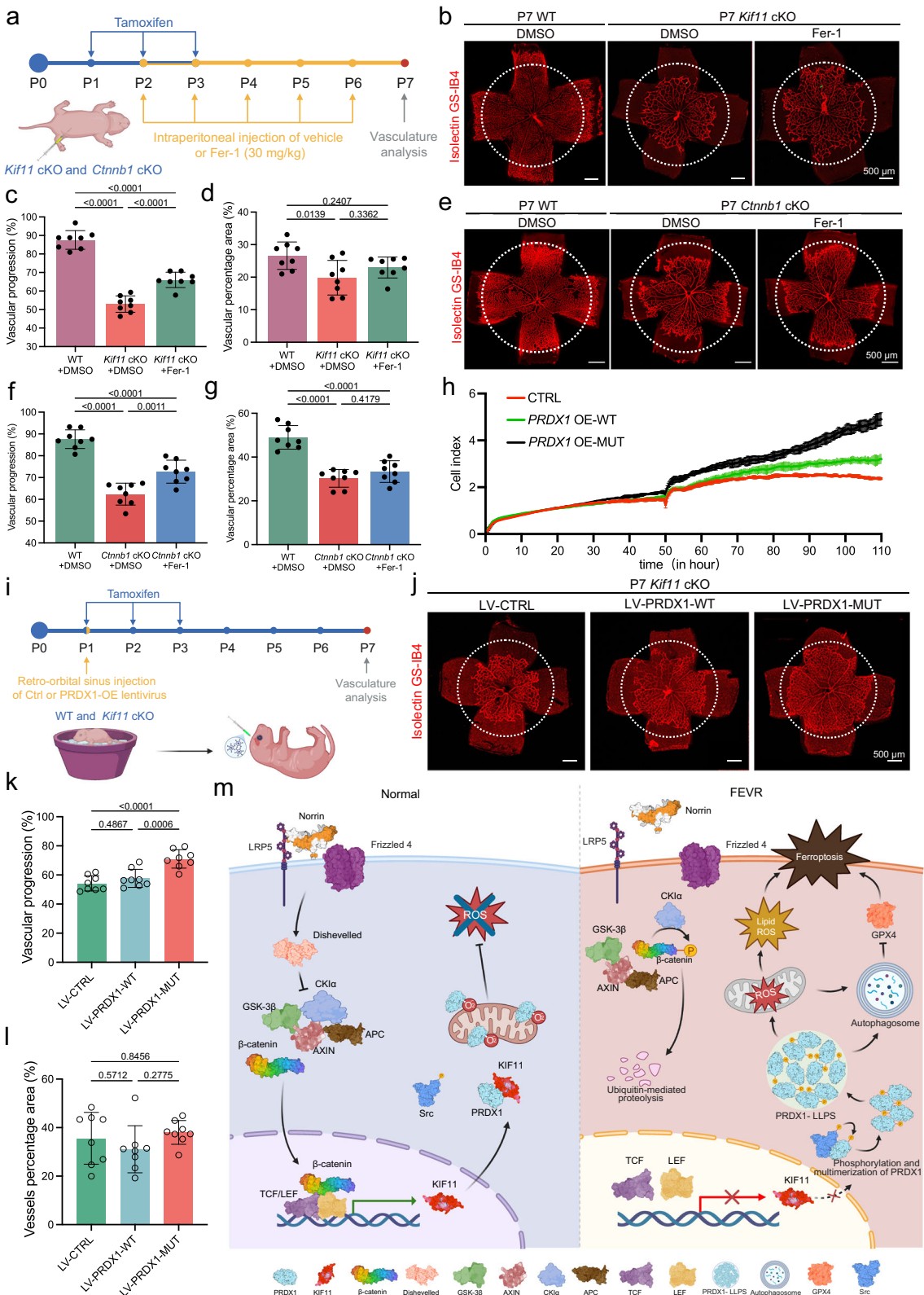

**LC-MS/MS targeted analysis of ferroptosis-related phospholipid peroxidation**

HRECs depleted with *KIF11* were collected for lipid extraction, which were subsequently subjected to targeted LC-MS/MS analysis of ferroptosis-related phospholipid peroxidation[11,33]. For lipid extraction, the collected HRECs were resuspended in 1 mL of a solvent mixture

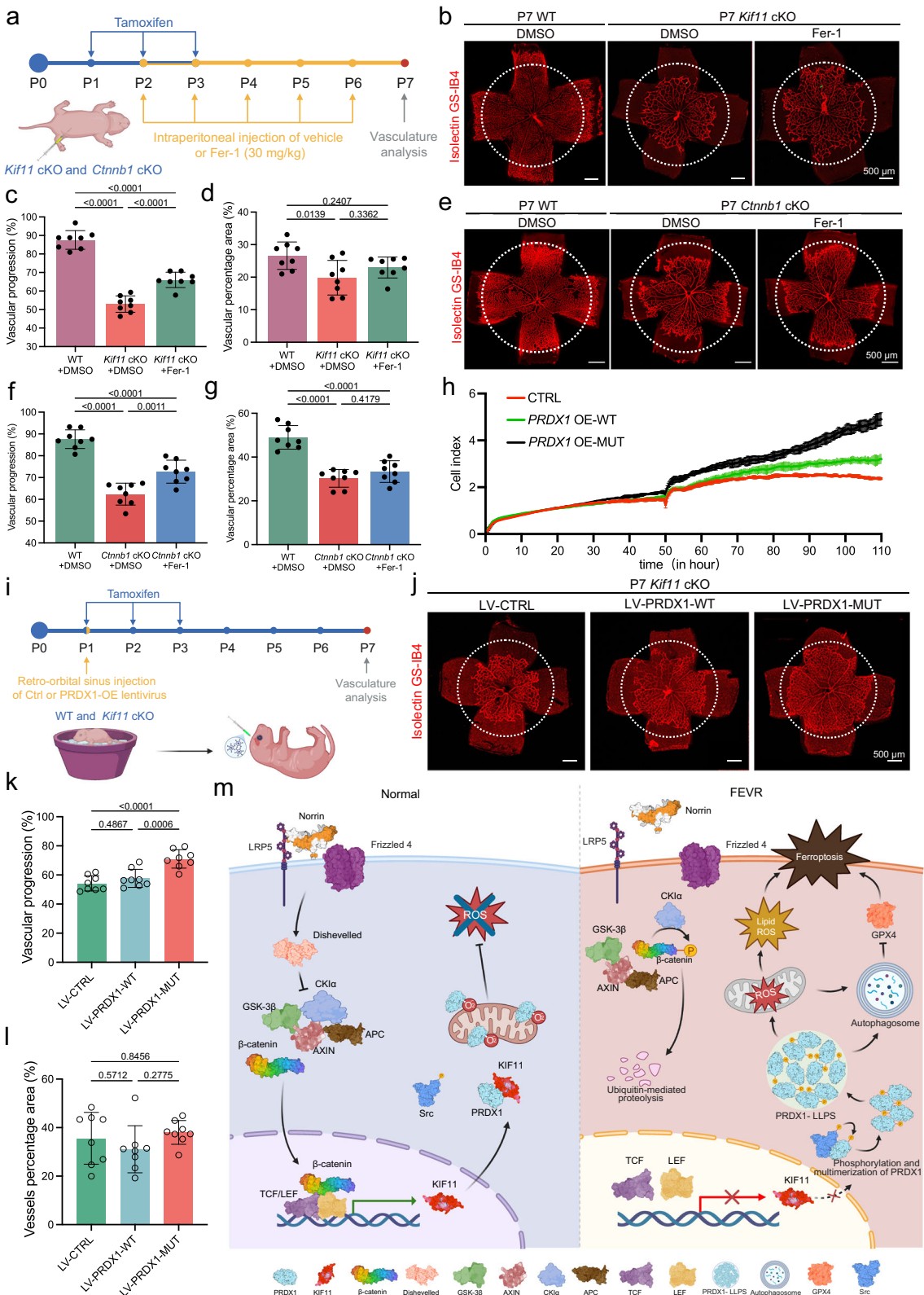

sample group, while all data in the other sample group are entirely missing. Then, the missing values of both groups were filled with the protein minimum and 1/2 the sample minimum, respectively. The differential analysis was performed using the Limma (version 3.40.6) package. Proteins with a $|\log_2 \text{FoldChange}| > 0.585$ and $p$-value $< 0.05$ were classified as differentially expressed.

**LC-MS/MS targeted analysis of ferroptosis-related phospholipid peroxidation**

HRECs depleted with *KIF11* were collected for lipid extraction, which were subsequently subjected to targeted LC-MS/MS analysis of ferroptosis-related phospholipid peroxidation[11,33]. For lipid extraction, the collected HRECs were resuspended in 1 mL of a solvent mixture

**Fig. 9 | Inhibiting ferroptosis or restoration of PRDX1 function partially rescues retinal vascular defects. a** Schematic diagram of the Fer-1 treatment design. Created in BioRender. Li, S. (2026) https://BioRender.com/b75j366. Representative retinal flat mounts (**b**) and quantification of retinal vascular progression (**c**) and vessel density (**d**) in P7 WT and *Kif11* cKO mice treated with DMSO, and *Kif11* cKO mice treated with Fer-1 (30 mg/kg). Circles indicate vessel outgrowth in *Kif11* cKO mice treated with Fer-1. Red, Isolectin GS-IB4. Scale bars, 500 µm. *n* = 8. Representative retinal flat mounts (**e**) quantification of retinal vascular progression (**f**) and vessel density (**g**) in P7 WT and *Ctnnb1* cKO mice treated with DMSO, and *Ctnnb1* cKO mice treated with Fer-1 (30 mg/kg). Circles indicate vessel outgrowth in *Ctnnb1* cKO mice treated with Fer-1. Red, Isolectin GS-IB4. Scale bars, 500 µm. *n* = 8. **h** Representative cell index values of CTRL, WT-, or Y194Q-mutant PRDX1 overexpressing HRECs. *n* = 3. **i** Schematic diagram of the rescue design of PRDX1-expressing lentivirus. Created in BioRender. Li, S. (2026) https://BioRender.com/b75j366. Representative retinal flat mounts (**j**) and quantification of retinal vascular

progression (**k**) and vessel density (**l**) in P7 *Kif11* cKO mice treated with LV-CTRL, LV-PRDX1-WT, or LV-PRDX1-MUT. Circles indicate vessel outgrowth in the *Kif11* cKO mice treated with LV-PRDX1-MUT. Red, Isolectin GS-IB4. Scale bars, 500 µm. *n* = 8. **m** Schematic of FEVR pathogenesis driven by β-catenin/KIF11 axis dysfunction. In normal retinal development, Norrin/β-catenin signaling induces KIF11, which competitively binds PRDX1 to inhibit its Src-mediated phosphorylation and subsequent liquid-liquid phase separation (LLPS). Conversely, FEVR-associated mutations in this pathway trigger the loss of KIF11 function and KIF11-PRDX1 interaction, leading to aberrant PRDX1 phosphorylation and LLPS. This protein condensation triggers ferroptosis, ultimately impairing retinal vascularization. Created in BioRender. Li, S. (2026) https://BioRender.com/b75j366. Data are presented as mean ± SD. *n* represents independent biological replicates or the number of mice per group. Statistical significance was determined using one-way ANOVA with Tukey's multiple comparisons test (**c, d, f, g, k, l**). Source data are provided as a Source Data file.

containing methanol, tert-butyl methyl ether (MTBE), butylated hydroxytoluene (BHT), along with internal standards. Lipid extracts were separated on a Thermo Accucore C30 column (2.6 µm, 2.1 × 100 mm) using the following gradient at 0.35 mL/min: 0 min (80% A, 20% B), 2 min (70% A, 30% B), 4 min (40% A, 60% B), 9 min (15% A, 85% B), 14 min (10% A, 90% B), 15.5 min (5% A, 95% B), 17.3 min (5% A, 95% B), and 17.5 min (80% A, 20% B). Mobile phase A consisted of acetonitrile/water (60:40, v/v) with 0.1% formic acid and 10 mM ammonium formate, while mobile phase B was acetonitrile/isopropanol (10:90, v/v) with the same additives. MRM mode was employed for detection on a 6500 Q-Trap mass spectrometer (Sciex, UK), monitoring parent-to-daughter ion transitions with the following settings: TEM 500°C, GS1 45, GS2 55, CUR 35, IS -4500V, DP -50V, EP -10V, and CE -38V. The peak area of each transition was integrated and normalized to the corresponding internal standard for quantification of lipid species. All lipid species analyzed had been previously identified and their structures confirmed by MS/MS[33,67].

### Co-immunoprecipitation (Co-IP) assay

The HRECs and HEK293T cells overexpressed with KIF11-FLAG were lysed with 300 µl cold lysis buffer for 20 min at 4 °C. For proximity-labeling, HEK293T co-overexpressing Src-miniTurboID-FLAG or Vec-miniTurboID-FLAG, KIF11-FLAG or Vector-FLAG, and PRDX1-HA were incubated with biotin at a final concentration of 500 µM for 30 min at 37 °C. After labeling, cells were immediately placed on ice and washed 3× with ice-cold PBS to remove free biotin. Then the cells were lysed with 300 µl cold lysis buffer for 20 min at 4 °C. After centrifugation, the soluble supernatants were incubated overnight at 4 °C with anti-FLAG Magnetic Beads for the KIF11-FLAG samples, or Streptavidin Magnetic Beads for the miniTurboID-based proximity-labeling (HY-K0207 and HY-K0208, Med Chem Express). The beads were washed three times with lysis buffer and resuspended in loading buffer before being heated for 5 min at 95 °C. The supernatants were loaded onto SDS-PAGE and processed for Western blot analysis. The antibodies used for western blot analysis were listed in Supplementary Table 5.

### Transmission electron microscopy (TEM)

The eyeballs and HRECs were prefixed with 3% glutaraldehyde, followed by postfixed with 1% osmium tetroxide, dehydrated in acetone, and embedded in Epon 812. Semithin sections were stained with toluidine blue for optical localization, while ultrathin sections were prepared using a diamond knife and stained with uranyl acetate and lead citrate. Observations were conducted using a JEM-1400 FLASH transmission electron microscope.

### Retinal leakage evaluation and immunofluorescence

For leakage evaluation, 100 µl EZ-Link™ Sulfo-NHS-Biotin (2 mg/ml, 21217, Thermo Fisher Scientific) was intraperitoneally injected into P7

mice 1 h before sacrifice. The retinal flat mounts were prepared and processed for immunofluorescence staining as described previously[3,4]. Briefly, the retinas were isolated from 4% FPA-fixed eyeballs and then fixed in methanol at −20 °C for 20 min, followed by rinsing with PBS 3 times and blocking with 5% donkey serum for 30 min at room temperature. The retinas were then incubated overnight at 4 °C with Isolectin GS-IB4 (I21413 or I21411, Invitrogen) and primary antibody, followed by 3 washes and incubation with secondary antibody or Streptavidin, Alexa Fluor™ 488 Conjugate (S11223, Thermo Fisher Scientific). The retinas were then flat-mounted using Fluoromount™ Aqueous Mounting Medium (F4680, Merck). The vascular progression and vessel density were measured using Zen Software (blue edition, version 2.6; Carl Zeiss) and Angiotool (version 0.5 beta). The images were obtained using a Zeiss LSM900 confocal microscope, Zeiss Axio Observer 7, and processed with Adobe Illustrator and Adobe Photoshop.

### Statistical and reproducibility

Statistical analyses were performed using GraphPad Prism 9.0. For comparisons between two groups, unpaired parametric Student's *t* tests were applied if the data passed normality and lognormality tests; otherwise, unpaired nonparametric tests with Welch's correction were used. For comparisons involving more than two groups and not otherwise specified in the figure legend, one-way or two-way ANOVA followed by Dunnett's or Tukey's multiple-comparison tests was conducted. Data are shown as mean ± SD unless otherwise specified. All experiments were independently repeated at least three times.

### Ethics approval and consent to participate

The clinical study complied with the Declaration of Helsinki and received approval from the ethical review boards of Sichuan Provincial People's Hospital (No.465, 2021) and Xinhua Hospital, Shanghai Jiaotong University (XHEC-KJB-2024-013). All animal experiment procedures were approved by the Animal Protection and Use Committee of Sichuan Provincial People's Hospital and were conducted in accordance with ethical standards, following the guidelines set by the Chinese Animal Welfare Committee for the care and use of laboratory animals.

### Reporting summary

Further information on research design is available in the Nature Portfolio Reporting Summary linked to this article.

## Data availability

The data supporting the findings from this study are available within the manuscript and its supplementary information. The raw targeted sequencing data generated in this study have been deposited in the

Genome Sequence Archive in the National Genomics Data Center[68], China National Center for Bioinformation, Beijing Institute of Genomics, Chinese Academy of Sciences, under accession number HRA009047. Researchers wishing to access the data must submit a formal request to the corresponding author, Shujin Li (lishujin91@126.com). The request should include a brief research proposal and a signed Data Use Agreement (DUA). We will respond to all access requests within 4 weeks. The data are restricted to academic use only and must not be used for re-identification of participants or shared with third parties without prior authorization. The raw data of bulk RNA-seq from *CTNNB1-*, *FZD4-*, *LRP5-*, *TSPAN12-*, *KIF11-*, and *PRDX1*-depleted HRECs, along with control HRECs, have been deposited in the Sequence Read Archive (SRA; PRJNA1297355, https://www.ncbi.nlm.nih.gov/bioproject/PRJNA1297355). The processed data for RNA-seq have been deposited in GEO (GSE306946, https://www.ncbi.nlm.nih.gov/geo/query/acc.cgi?acc=GSE306946). FPKM values for bulk RNA-seq were listed in Supplementary Data 1. The raw data of scRNA-seq from P6 WT, P6 *Tspan12* KO, and P6 *Ctnnb1* cKO mice have been deposited in the GEO database (GSE305756, https://www.ncbi.nlm.nih.gov/geo/query/acc.cgi?acc=GSE305756). The mass spectrometry proteomics data of KIF11-binding proteins and *KIF11*-depleted HRECs have been deposited in the ProteomeXchange Consortium (https://www.ebi.ac.uk/pride/archive/projects) via the PRIDE partner repository[69], with the dataset identifiers PXD074452 and PXD074456, respectively. The processed data of targeted phospholipid peroxidation have been deposited to MetaboLights (https://www.ebi.ac.uk/metabolights/MTBLS12912). Source data are provided with this paper.

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

## Acknowledgements

We thank all the patients and their family members for participating in this study. This research is supported by the Development Center for Medical Science and Technology, National Health Commission (2024ZD0523901), the Science Fund for Creative Research Groups of the National Natural Science Foundation of China (82121003), the National Natural Science Foundation of China (82471102, 82571242, 82401292, 82330030, and 82301231), the Joint Funds of the National Natural Science Foundation of China (U25A6002), the National Key R&D Program of China (2024YFC2510900), the Sichuan Science and Technology program (2023ZYD0172), the Jinfeng Laboratory (JFLKYXM202403AZ-101), the China Postdoctoral Science Foundation (2024M750365 and 2023M740518), the Fundamental Research Funds for the Central Universities of Ministry of Education of China (ZYGX2022J023), and the Research Fund of Sichuan Academy of Medical Sciences and Sichuan Provincial People's Hospital (2023BH09).

## Author contributions

Z.Y., S.L., and M.Y. co-conceived the study and wrote the manuscript. M.Y., R.Z., L.P., and L.L. conducted the experiments and data analysis. L.P. helped with the retro-orbital venous sinus injection. L.Y. and H.X. helped with mouse breeding. P.Z. and X.Z. prepared the clinical samples. Y.H. and X.H. performed Sanger sequencing on the clinical samples.

## Competing interests

The authors declare no competing interests.

## Additional information

[1]Genetic Diseases Key Laboratory of Sichuan Province, Department of Medical Genetics, Sichuan Academy of Medical Sciences & Sichuan Provincial People's Hospital, School of Medicine, University of Electronic Science and Technology of China, Chengdu, PR China. [2]Sichuan-Chongqing Joint Key Laboratory of Pathology and Laboratory Medicine, Jinfeng Laboratory, Chongqing, PR China. [3]Department of Ophthalmology, Sichuan Academy of Medical Sciences & Sichuan Provincial People's Hospital, School of Medicine, University of Electronic Science and Technology of China, Chengdu, PR China. [4]Department of Ophthalmology, Xin Hua Hospital Affiliated to Shanghai Jiao Tong University School of Medicine, Shanghai, PR China. [5]Research Unit for Blindness Prevention, Chinese Academy of Medical Sciences (2019RU026), Sichuan Academy of Medical Sciences & Sichuan Provincial People's Hospital, Chengdu, PR China. [6]These authors contributed equally: Mu Yang, Rulian Zhao, Li Peng, Liting Lv. [7]These authors jointly supervised this work: Shujin Li, Zhenglin Yang. ✉e-mail: lishujin91@126.com; yangzhenglin@cashq.ac.cn

