## [Transparent Peer Review File · Nature Communications]

KIF11 prevents retinal endothelial ferroptosis in familial exudative vitreoretinopathy by inhibiting phosphorylation-driven PRDX1 phase separation

Corresponding Author: Professor Shujin Li

Version 0:

Reviewer comments:

Reviewer #1

(Remarks to the Author)

Yang et al. report that KIF11 is a downstream target of beta-catenin signaling in HRMVECs and regulates the activity of the antioxidant protein Peroxiredoxin 1 by a physical interaction that promotes T194 PRDX1 phosphorylation and LLPS. Kif11 or PRDX1 or CTNNB1 deficiency all induce ferroptosis in HRMVECs. In mice, Kif11 endothelial KO angiogenesis phenotypes are restored by the ferroptosis inhibitor ferrostatin or viral overexpression of non-phosphorylated PRDX1. Conceptually, there is major progress in the model that KIF11 is downstream of beta-catenin signaling because it is not known why KIF11 is a FEVR gene and if it has a role within the norrin/frizzled4 pathway. There is also a major advance in the model that KIF11 regulates PRDX1 and ferroptosis, with implications for FEVR. This is a major study based on the novelty of the findings and models. This is also a massive set of data, generally with appropriate controls and quantifications. Therefore, this reviewer will focus on a few key issues.

The claim that KIF11 is a downstream target of beta-catenin dependent signaling.

- This claim will be controversial, because scRNAseq data from Ndp KO and Tspan12 KO mice did not identify KIF11 as a downstream target of beta-catenin signaling (Heng et al 2019 (Nathans lab), Zhang et al 2023 (Junge lab)). In addition, transcriptomics in HRMVECS and other CNS EC cell lines +/- norrin stimulation was conducted by multiple groups and there is no report that Kif11 is a downstream target. This could be due to differences in developmental timing, i.e., maybe KIF11 is a target of norrin signaling only in developing (non-quiescent) endothelial cells. This is a key claim of the study, because it links ferroptosis to loss of beta-catenin signaling and FEVR. Therefore, the authors should examine one of the KO strains in which mediators of the norrin/frizzled4 pathway are gene ablated (e.g., Ndp KO mice or Fzd4 KO mice) and confirm that KIF11 is downregulated in the ECs of these mice and that a ferroptosis inhibitor alleviates phenotypes (at least some of the phenotypes) in these mice. This experiment is critical to substantiate the model in this manuscript.

- In addition, to rule out that Kif11 is dysregulated as response to lentiviral infection or as off-target effect of the shRNA, the qPCR data in Figure 2A should be repeated with an independent siRNA (which targets a different region of the target), without using lentivirus for delivery.

Weaknesses in the analyses of the transcriptomic data.

The following can be addressed without new experiments:

- The authors should highlight and discuss that the knockdown of FZD4, LRP5, TSPAN12, and CTNNB1 was apparently done in the absence of norrin stimulation. This is important because target genes of beta-catenin-dependent signaling may not be affected by the KD in the absence of norrin, and neither HRMVECs nor serum are a source of norrin.

- The supplemental transcriptomic data in the excel file are not readily accessible because the accession numbers but no gene names are provided. This makes it difficult to critically evaluate the data, as every gene of interest has to be translated into an accession number before the gene can be looked up in this data.

- A clarification is lacking whether the same control samples were used for all KD comparisons, or if different control samples

were used for each KD experiment (apparently the latter based on variable RPK in sets of control samples).

- The data don't look good with respect to consistency across CTNNB1, FZD4, TSPAN12, and LRP5 KD for key genes, despite the claim in Fig 1C that there are consistent differences. Note also that in Fig 1B there is little to no overlap in the top upregulated and top downregulated genes, this should be discussed in the manuscript.

- There is no discussion on what genes are expected to be regulated by beta-catenin signaling (e.g., Axin2 and Apcdd1), and how these markers of Wnt signaling change across the various KD conditions. Contrary to expectations, Axin2 is upregulated in Tspan12 KD cells, and it is not downregulated in CTNNB1 KD or LRP5 KD cells. Also contrary to expectations, APCDD1 is upregulated in Tspan12 KD cells. Together, the evidence is not strong that knock down conditions would affect beta-catenin signaling.

There is no discussion in the manuscript of how many fold KIF11 was downregulated in each KD experiment, if the change was significant (no statistics shown for the RNAseq in the supplemental table), and how Kif11 ranks relative to other downregulated genes in each knock down condition (is it in the top10? top 100?)

- It is unclear why the authors selected the dark magenta module, was the negative correlation with control and the trend towards a positive correlation with knock down stronger than for any of the other 23 gene modules? This is not described, was the module selected based on subjective interest?

In vivo experiments

- Figure 10 p and 10 t. Clarify in the manuscript if littermates were used. Add the control group (Cre-) for the qualitative and quantitative data so the extent of rescue vs this control can be quantified.

- Unclear how a retroorbital viral delivery in mouse pups works (with closed eyelids at this stage), and how effective it is. An experiment with a GFP virus can show how many ECs are transfected and validate this delivery method at this age.

Minor:

- some of the fluorescent cell images are too dim (e.g., Hoechst in Fig 8h and multiple others)
- Fig 6K should show the full native gel so that all MW specimens of PRDX1 can be seen

Reviewer #2

(Remarks to the Author)

In the presented manuscript, Yang, Zhao Peng and Lv et al. discovered KIF11 as a novel downstream target of the Norrin/beta-catenin pathway in HRECs in a screen. Mutations in KIF11 have been previously reported in the context of human eye diseases. Besides their models of endothelial cells of the eyes and mouse retinas (e.g. Fig. 9a), the authors might want to step a little further and test KIF11 in general models of ferroptosis, standard cells and assays that might expand their discovery from eye-specific to generally important to ferroptosis. I therefore suggest a couple of cell culture experiments in my list of major concerns below. Once this is performed, the manuscript will be of interest not only to eye specialists, but also to the broader readership of a journal such as Nature Communications.

In summary, at the current stage, the manuscript is quite specialized, but I can envision a broader role for their discovery in ferroptosis which I recommend to be tested in some detail.

Major concerns

- What is the role of KIF-11 in ferroptosis, if any? This is easy to investigate by knocking out and back in AKR1 in standard ferroptosis cell lines (HT1080, Pfa1) treated with various ferroptosis inducers of all classes (erastin, RSL3, FIN56, FINO2). As readout systems, time lapse imaging, PI staining, LDH release, any other vital dye, annexin V etc should be investigated?
- What is the baseline expression and overall relevance of the reported Kif-11-system? Western blots should be provided with established (ko-controlled) antibodies. How does the absence of KIF-11 relate to the known ferroptosis-controlling systems such as GPX4, FSP1 etc.? How does it affect the baseline levels of hydroperoxides, ether lipids etc?
- How does KIF-11 regulate ferroptosis? This requires the monitoring of cellular systems (HT1080, NIH3T3, Pfa1 and potentially the cells used in the screen in Fig. 1) over time. How plastic are the above-mentioned systems once induced in the absence of KIF-11? How does the peroxidative lipid profile assessed by LC/MS change over time? How does this phenocopy MALDI-imaging of the corresponding lipids in section of these eyes?
- Fig. 7-9 are a strong part of this manuscript. But which kinase is responsible for the LLPS and PRDX1 phosphorylation, and how does KIF-11 regulate it?
- The authors conclude that autophagy dependent ferroptosis is key here. However, they do not demonstrate the requirement of the "autophagy" part. Therefore, I highly recommend to reduce this controversial term "ferroptosis" instead of "autophagy-dependent ferroptosis". As with other regulated cell death pathways, autophagy is triggered mostly in parallel to the cell death pathways of most kinds. In that sense, autophagy is rather a counterregulatory attempt of the cell to improve its energy profile and may be interpreted as a rescue attempt, also during the process of ferroptosis.

Reviewer #3

(Remarks to the Author)

The overall study has positive significance for the mechanistic research of KIF11. The authors conclude that: modulation of the Norrin pathway induces changes in KIF11; KIF11 regulates alterations in the autophagy and ferroptosis pathways; KIF11

interacts with PRDX1, and the loss of the KIF11/PRDX1 interaction leads to PRDX1 polymerization and triggers liquid-liquid phase separation (LLPS). Additionally, ferroptosis inhibitors and PRDX1 overexpression can partially rescue the vascular developmental defects caused by KIF11.

However, this study has several notable limitations:

Logical Inconsistencies: The experimental design is fragmented, with many proposed connections relying excessively on bioinformatic correlations. The authors have not established a complete mechanistic chain through functional experiments, and some of the bioinformatic analyses appear redundant.

Weak Evidence for Autophagy-Dependent Ferroptosis: Although KIF11 is shown to induce autophagy and ferroptosis separately, there is insufficient experimental evidence to definitively demonstrate that KIF11 triggers ferroptosis through autophagy-dependent mechanisms. The reliance on a single shRNA for knockdown experiments undermines the reliability of these findings.

Abrupt Presentation of KIF11 Mutation Data: The inclusion of KIF11 mutation results lacks contextual integration with the overall narrative, appearing disconnected from the core mechanistic exploration.

Incomplete Animal Experiments: The methodology for drug treatment in animal models is inadequately described.

Phenotypic analysis is superficial, focusing primarily on vascular developmental status while neglecting critical assessments (e.g., blood-retinal barrier integrity). Given the close relationship between vascular development and overall growth in mice, the authors should provide body weight statistics for all experimental groups in the supplementary materials.

Version 1:

Reviewer comments:

Reviewer #1

(Remarks to the Author)

The authors conducted additional experiments that strengthen the conclusions of the study. With respect to the claim that KIF11 is a target gene of norrin signaling, the authors focussed on proliferating ECs and found a reduction of KIF11 mRNA in P6 proliferating ECs in Tspan12 and Ctnnb1 mutant cells (Fig. 2h, statistics not described in the figure legend). Lentivirus mediated delivery of KIF11 partially rescued retinal vascular outgrowth in Ctnnb1 mutant retinas. Whether norrin induces KIF11 in HRMVECs remains unclear. Together, concerns raised by this reviewer were sufficiently addressed.

Reviewer #2

(Remarks to the Author)

The authors have greatly improved their manuscript. However, I recommend to test one more important control in the HT1080 cells (Supplementary Figure 8). Would the cell death observed upon knockdown of KIF11 be reversed by the addition of a ferroptosis inhibitor (e.g. Fer-1)?

If the authors cannot provide a crCas9-mediated knockout of Kif-11 in HT1080 cells, they should at least discuss their limitations (PMID38489367).

Otherwise, I recommend this manuscript for publication!

Reviewer #3

(Remarks to the Author)

The manuscript has significantly improved in this revised version. Both the experimental design and the writing are more logically structured than before. The authors have incorporated an additional shRNA to assess their key conclusion, and they have reorganized the section discussing the relationship between KIF11 and autophagy in conjunction with ferroptosis. Moreover, the mechanisms underlying how the LoF of KIF11 impairs endothelial cells through interaction with PRDX1 in the context of FEVR are now well addressed. However, there are several minor issues that still require refinement.

1. Lines 246-247: The authors conclude, "Taken together, these results suggest that the loss of KIF11 might trigger autophagy during retinal vascular development." However, in the in vitro experiments and bioinformatics analysis involving Primary Human Retinal Microvascular Endothelial Cells, the donor's age is not provided on the official website of the primary cell line. Therefore, it is imprecise to attribute this phenotype to "retinal vascular development." This conclusion would be more appropriate in the context of the in vivo results.

2. Lines 278-279: The statement regarding the protective effect of KIF11 against ferroptosis is inaccurate, as no experimental data regarding the overexpression of KIF11 to mitigate ferroptosis are presented in this section.

3. The MWs of PRDX1's LMW and HWM forms should be mentioned, or at least an approximate scale should be provided.

Version 2:

Reviewer comments:

Reviewer #2

(Remarks to the Author)

The authors have adequately responded to all concerns raised by this referee, but still, I believe that a crCas9-based knockout is required for the conclusions. An shRNA or an siRNA in both cases may affect ferroptosis. However, this is discussed as a limitation of this work, so I recommend to publish the manuscript.

Response to Reviewer's Comments

General response to reviewers' comments:

We thank the reviewers for their thorough and constructive feedback, which guided us to conduct multiple new experiments over the past few months, resulting in a more comprehensive mechanistic framework. The major advancements include:

1. Incorporation of scRNA-seq data: We have performed scRNA-seq on retinal vascular cells from *Ctnnb1* cKO and *Tspan12* KO mice. Integrated with public datasets and validated by additional experiments, these data confirmed that *Kif11* is specifically expressed in proliferative ECs of postnatal retinas, but largely absent in adult retinas, and is downregulated in *Ctnnb1* cKO and *Tspan12* KO retinal ECs. These findings strengthen the conclusion that *KIF11* is regulated by β -catenin signaling and explain the early-onset nature of *KIF11*-related FEVR.

2. Ferroptosis-related experiments: In HRECs, a combination of assays—including time-lapse live-cell imaging, LDH release, and LC-MS/MS phospholipid peroxidation profiling—demonstrated that *KIF11* knockdown induces ferroptosis. Parallel studies in HT1080 cells yielded consistent results, suggesting a broader role of KIF11 in ferroptosis regulation beyond endothelial cells.

3. Mechanistic studies on PRDX1 phosphorylation: Western blot and inhibitor assays confirmed that PRDX1 phosphorylation in *KIF11*-depleted HRECs is mediated by Src kinase. miniTurboID-based Co-IP further revealed that KIF11 disrupts the Src-PRDX1 interaction, likely through spatial occupancy of the Src-binding site, thereby preventing Src-mediated phosphorylation.

4. *In vitro* and *in vivo* rescue experiments: We have incorporated additional rescue experiments. Fer-1 treatment exhibits partial rescue effects on *CTNNB1*-depleted HRECs and *Ctnnb1* cKO mice. In addition, retro-orbital sinus delivery of KIF11-expressing lentivirus partially restored retinal vascular progression in *Ctnnb1* cKO mice.

Conclusion: Together, these new results provide strong *in vitro* and *in vivo* evidence that the β -catenin/KIF11/PRDX1 axis plays a critical role in ferroptosis and retinal vascular development, highlighting the potential of targeting ferroptosis or restoring KIF11 function as therapeutic strategies for related retinal vascular disorders.

REVIEWER COMMENTS

Reviewer #1 (Remarks to the Author):

Yang et al. report that KIF11 is a downstream target of β -catenin signaling in HRMVECs and regulates the activity of the antioxidant protein Peroxiredoxin 1 by a physical interaction that promotes T194 PRDX1 phosphorylation and LLPS. Kif11 or PRDX1 or CTNNB1 deficiency all induce ferroptosis in HRMVECs. In mice, Kif11 endothelial KO angiogenesis phenotypes are restored by the ferroptosis inhibitor ferrostatin or viral overexpression of non-phosphorylated PRDX1. Conceptually, there is major progress in the model that KIF11 is downstream of β -catenin signaling because it is not known why KIF11 is a FEVR gene and if it has a role within the norrin/frizzled4 pathway. There is also a major advance in the model that KIF11 regulates PRDX1 and ferroptosis, with implications for FEVR. This is a major study based on the novelty of the findings and models. This is also a massive set of data, generally with appropriate controls and quantifications. Therefore, this reviewer will focus on a few key issues.

1. The claim that KIF11 is a downstream target of β -catenin dependent signaling.

- This claim will be controversial, because scRNAseq data from Ndp KO and Tspan12 KO mice did not identify KIF11 as a downstream target of β -catenin signaling (Heng et al 2019 (Nathans lab), Zhang et al 2023 (Junge lab)). In addition, transcriptomics in HRMVECS and other CNS EC cell lines +/- norrin stimulation was conducted by multiple groups and there is no report that Kif11 is a downstream target. This could be due to differences in developmental timing, i.e., maybe KIF11 is a target of norrin signaling only in developing (non-quiescent) endothelial cells. This is a key claim of the study, because it links ferroptosis to loss of β -catenin signaling and FEVR. Therefore, the authors should examine one of the KO strains in which mediators of the norrin/frizzled4 pathway are gene ablated (e.g., Ndp KO mice or Fzd4 KO mice) and confirm that KIF11 is downregulated in the ECs of these mice and that a ferroptosis inhibitor alleviates phenotypes (at least some of the phenotypes) in these mice. This experiment is critical to substantiate the model in this manuscript.

Response:

We sincerely appreciate the reviewer's insightful comments and valuable suggestions. To address this controversial claim, we have incorporated data from endothelial cell-specific *Ctnnb1* (encode β -catenin) conditional knockout (*Ctnnb1* cKO) and *Tspan12*

(encode TSPAN12) global knockout (*Tspan12* KO) mice, and have performed single-cell RNA sequencing (scRNA-seq) on P6 retinal vascular cells isolated from both models. Furthermore, as mentioned by the reviewer, we have integrated the relevant public datasets (P60 WT and *Ndp* KO from Heng et al 2019 of Nathans lab; P26 WT and *Tspan12* KO from Zhang et al 2023 of Junge lab) with our data.

Consistent with the interpretation proposed by the reviewer that *KIF11* might be a target of Norrin/ β -catenin signaling only in developing (non-quiescent) ECs, the integrated scRNA-seq analysis confirmed that *Kif11* is predominantly expressed in proliferative ECs of postnatal retinas (non-quiescent), but largely absent in adult retinas (quiescent, P26 and P60) (Fig. 2d-h and Supplementary Fig. 2d, e). We have also incorporated western blot and RT-qPCR analyses on P1, P7, P14, P21, and P28 retinas and confirmed the marked reduction of KIF11 expression after P7 (Fig. 2i and Supplementary Fig. 2g). Therefore, it is not applicable to test whether *KIF11* is altered in quiescent retinal ECs from adult mice with defective Norrin/ β -catenin signaling.

In the postnatal retinal ECs, scRNA-seq revealed significant downregulation of *Kif11* in total EC (Fig. 2d) and proliferative EC (Fig. 2h) of *Ctnnb1* cKO and *Tspan12* KO groups compared with the WT. The raw scRNA-seq data have been deposited in the NCBI Gene Expression Omnibus (GEO) database under accession GSE305756.

Additionally, to further test the role of KIF11 in postnatal vascular development in the absence of endothelial β -catenin, we have applied retro-orbital injection of KIF11-expressing lentivirus, which resulted in specific expression of KIF11 in retinal vascular cells and partially rescued vascular progression in *Ctnnb1* cKO mice (Fig. 2p). These findings support the conclusion that KIF11 functions downstream, at least in part, of Norrin/ β -catenin signaling to promote retinal vascular development.

Furthermore, in response to the reviewer's suggestion, we have conducted rescue experiments in *Ctnnb1* cKO mice using Fer-1 treatment. The results demonstrated partial rescue effects on the retinal vascular progression of *Ctnnb1* cKO (Fig. 9e, f), suggesting that ferroptosis contributes to the vascular defects observed in FEVR.

2. In addition, to rule out that *Kif11* is dysregulated as response to lentiviral infection or as off-target effect of the shRNA, the qPCR data in Figure 2A should be repeated with an independent siRNA (which targets a different region of the target), without using lentivirus for delivery.

Response:

We appreciate the reviewer's thoughtful suggestion. To rule out the possibility that the downregulation of *KIF11* results from lentiviral infection or off-target effects of the original shRNA, we have conducted knockdown experiments in HRECs using an independent shRNA targeting a different region of *CTNNB1*, *FZD4*, *LRP5*, or *TSPAN12*. These experiments reproduced the reduction in *KIF11* expression (Supplementary Fig. 1e-h).

Additionally, the reviewer mentioned delivering siRNA without using lentivirus. However, transient transfection in HRECs does not achieve high delivery efficiency, as has been reported previously¹. To rule out potential non-specific effects of viral infection on HRECs, the lentivirus-delivered non-targeting shRNA was used as the control. Taken together, these results indicate that the observed *KIF11* downregulation following *CTNNB1*, *FZD4*, *LRP5*, or *TSPAN12* knockdown is not attributable to viral infection or off-target effects of the original construct.

Weaknesses in the analyses of the transcriptomic data.

The following can be addressed without new experiments:

3. The authors should highlight and discuss that the knockdown of *FZD4*, *LRP5*, *TSPAN12*, and *CTNNB1* was apparently done in the absence of norrin stimulation. This is important because target genes of β -catenin-dependent signaling may not be affected by the KD in the absence of norrin, and neither HRMVECs nor serum are a source of norrin.

Response:

We thank the reviewer for this comment. We agree with the reviewer that, although our RNA-seq and further analyses identifying *KIF11* as a hub gene downregulated upon knockdown of *FZD4*, *LRP5*, *TSPAN12*, or *CTNNB1* were originally performed without Norrin stimulation, it is indeed important to assess *KIF11* expression in the presence of Norrin. To address this, we have incorporated RT-qPCR analysis with recombinant Norrin protein treatment (50 ng/ml for 24 h). Consistent with the findings obtained in the absence of Norrin, *KIF11* was downregulated by more than two-fold upon knockdown of *FZD4*, *LRP5*, *TSPAN12*, or *CTNNB1* in the presence of Norrin (Supplementary Fig. 1i). The related description has been added in the main text. Therefore, *KIF11* expression was reduced upon silencing of core components of Norrin/ β -catenin signaling, regardless of Norrin stimulation.

4. The supplemental transcriptomic data in the excel file are not readily accessible

because the accession numbers but no gene names are provided. This makes it difficult to critically evaluate the data, as every gene of interest has to be translated into an accession number before the gene can be looked up in this data.

Response:

We thank the reviewer for pointing this out. To improve data accessibility, we have revised the supplemental transcriptomic data file to include gene symbols with FPKM values for each sample (Supplementary Table S1). The updated file has been uploaded with the revised submission. Additionally, the corresponding raw and processed data have been deposited in the NCBI Gene Expression Omnibus (GEO) database under accession GSE306946.

5. A clarification is lacking whether the same control samples were used for all KD comparisons, or if different control samples were used for each KD experiment (apparently the latter based on variable RPK in sets of control samples).

Response:

We thank the reviewer for raising this critical point. In the previous version of the manuscript, the different knockdown groups were compared with different control samples from separate batches, which could have introduced discrepancies in downstream analyses. Therefore, we have newly performed transcriptome sequencing for the knockdown of *FZD4*, *LRP5*, *TSPAN12*, and *CTNNB1* using the same control samples. The *KIF11* and *PRDX1* knockdown datasets were compared to their own corresponding non-targeting shRNA control. This clarification has been added to the Methods part.

6. The data don't look good with respect to consistency across *CTNNB1*, *FZD4*, *TSPAN12*, and *LRP5* KD for key genes, despite the claim in Fig 1C that there are consistent differences. Note also that in Fig 1B there is little to no overlap in the top upregulated and top downregulated genes, this should be discussed in the manuscript.

Response:

We thank the reviewer for this important suggestion. As mentioned above, we have performed RNA-seq for *FZD4*, *LRP5*, *TSPAN12*, and *CTNNB1* knockdown groups using the same control samples within a single experimental batch. Volcano plots of the newly generated bulk RNA-seq data revealed partial overlap among the top five upregulated and downregulated genes across the *CTNNB1*, *FZD4*, *TSPAN12*, or *LRP5* knockdown. Notably, *TNF* and *SMIM* ranked among the top 5 downregulated genes in three of four groups (Fig. 1b). Additionally, several classical Wnt downstream targets—including

PLVAP (upregulated) and *APCDD1*, *DKK1* and *SLC2A1* (downregulated)—have been incorporated in Fig. 1c and highlighted in Fig. 1b.

Furthermore, to assess the consistency of transcriptomic change following knockdown of *CTNNB1*, *FZD4*, *TSPAN12*, or *LRP5*, we have performed a Venn diagram analysis of the differentially expressed genes (DEGs) in each condition. This analysis revealed that approximately 25% of the DEGs ($|\log_2FC| > 0.585$, $FDR < 0.05$) were commonly altered across groups (Supplementary Fig. 1b-d), indicating a substantial degree of overlap and supporting the presence of a core transcriptomic response associated with disruption of the β -catenin pathway, even in the absence of Norrin. The \log_2FC values of each gene across all groups analyzed by Venn diagram have been provided in Supplementary Table 2. The corresponding description has been added to the main text.

7. There is no discussion on what genes are expected to be regulated by β -catenin signaling (e.g., *Axin2* and *Apcdd1*), and how these markers of Wnt signaling change across the various KD conditions. Contrary to expectations, *Axin2* is upregulated in *Tspan12* KD cells, and it is not downregulated in *CTNNB1* KD or *LRP5* KD cells. Also contrary to expectations, *APCDD1* is upregulated in *Tspan12* KD cells. Together, the evidence is not strong that knock down conditions would affect β -catenin signaling.

There is no discussion in the manuscript of how many fold *KIF11* was downregulated in each KD experiment, if the change was significant (no statistics shown for the RNAseq in the supplemental table), and how *Kif11* ranks relative to other downregulated genes in each knock down condition (is it in the top10? top 100?)

Response:

We are grateful to the reviewer for highlighting this important point. We have analyzed the newly generated RNA-seq data and found consistent regulation of several genes known to be regulated by the β -catenin pathway^{2,3,4}—including *APCDD1*, *DKK1*, *PLVAP*, and *SLC2A1*—across the various knockdown conditions. In line with previous findings, *APCDD1*, *DKK1*, and *SLC2A3* were consistently downregulated, while *PLVAP* was significantly upregulated following knockdown of either *CTNNB1*, *FZD4*, *TSPAN12*, or *LRP5* (Fig. 1b, c). The corresponding description has been added to the main text.

However, *AXIN2* showed only a slight, non-significant downregulation in the *CTNNB1* and *FZD4* knockdown groups (see the figure below), which may reflect variability due to its inherently low expression levels in HRECs (the average *AXIN2* FPKM level in control

group of HRECs is 0.5, compared with 51.9 and 5.5 of *DKK1* and *SLC2A1*, respectively). The FPKM values in each sample could be found in Supplementary Table S1.

As to the expression levels of *KIF11* in the bulk RNA-seq data, we have incorporated a dot-box plot of *KIF11* expression levels (FPKM value) across all groups in Fig. 1i, which revealed a significant ($FDR < 0.05$) reduction of *KIF11* upon knockdown of either *CTNNB1*, *FZD4*, *TSPAN12*, or *LRP5* with Log_2FC of -1.62, -0.92, -1.22, -1.23, respectively. The relevant description has been added to the figure legend.

In terms of ranking, *KIF11* was among the top 76 downregulated genes when using a more stringent Venn analysis parameter ($\text{log}_2\text{FC} < -0.91$, $FDR < 0.05$) across the knockdown conditions. The relevant description has been added to the main text. As mentioned in the response to comment 6, the Log_2FC values of each gene across all groups included in the Venn diagram have been provided in Supplementary Table S2.

Although *KIF11* was not among the most highly downregulated genes, we considered it a potential downstream target of the β -catenin pathway for two main reasons: (i) WGCNA followed by CytoHubba analysis identified *KIF11* as a top-five hub gene ranked by MCC (Maximal Clique Centrality) score, and (ii) *KIF11* has been previously identified as a FEVR-associated gene.

8. It is unclear why the authors selected the dark magenta module, was the negative correlation with control and the trend towards a positive correlation with knock down stronger than for any of the other 23 gene modules? This is not described, was the module selected based on subjective interest?

Response:

We appreciate the reviewer's thoughtful question regarding our selection of the dark magenta module. In the previous version, this module was selected because it showed a distinct and statistically significant correlation pattern across conditions, with the strongest negative correlation with the control group (ME = -0.96), compared with other modules such as midnight blue (ME = -0.77) and yellow (ME = -0.58). Therefore, the dark magenta module was selected for further analysis due to its robust association with knockdown-induced transcriptomic changes.

In the revised version, however, the blue module was selected, as it exhibited the strongest negative correlation with the control group (ME = -0.71). Notably, further enrichment analysis yielded results similar to those in the previous version (Fig. 1f).

In vivo experiments

9. Figure 10 p and 10 t. Clarify in the manuscript if littermates were used. Add the control group (Cre-) for the qualitative and quantitative data so the extent of rescue vs this control can be quantified.

Response:

We thank the reviewer for this comment. In the previous version, littermates were used for phenotypic comparisons of the mouse models, and the corresponding descriptions have been added to the manuscript. In the revised version, most of the newly added experiments were also performed using littermate controls. However, the scRNA-seq data for *Ctnnb1* cKO, *Tspan12* KO, and control samples were not obtained from littermates, as more than four mice are required to obtain sufficient retinal vascular cells for scRNA-seq, which exceeded the number available per litter.

In addition, we have added figures showing non-targeting lentivirus-treated littermate controls (Flox/Flox) for the KIF11-expressing lentivirus rescue experiments on *Ctnnb1* cKO mice (Fig. 2p-r), as well as DMSO-treated littermate controls for the Fer-1 rescue experiments on *Kif11* cKO (Fig. 9b-d) and *Ctnnb1* cKO mice (Fig. 9e-g). These analyses demonstrated partial rescue effects of KIF11-expressing lentivirus in *Ctnnb1* cKO mice (Fig. 2p-r) and of Fer-1 treatment in *Kif11* cKO (Fig. 9b-d) and *Ctnnb1* cKO mice (Fig. 9e-g) FEVR mice.

10. Unclear how a retroorbital viral delivery in mouse pups works (with closed eyelids at this stage), and how effective it is. An experiment with a GFP virus can show how many ECs are transfected and validate this delivery method at this age.

Response:

We thank the reviewer for pointing this out. Retro-orbital venous sinus injection is a widely used method for compound intravenous administration and vessel visualization in neonatal mice with closed eyelids^{5, 6}. As suggested, we have now included images of flat-mounted retinas from P7 mice that received retro-orbital sinus injections of GFP-expressing lentivirus at P1, alongside uninjected controls. Immunofluorescence confirmed robust GFP expression in the retinal vasculature, demonstrating efficient viral delivery and transduction (Supplementary Fig. 3j).

Minor:

11. some of the fluorescent cell images are too dim (e.g., Hoechst in Fig 8h and multiple others)

Response:

We thank the reviewer for this comment. In the revised version, we have optimized the imaging settings and replaced the representative images of Fig. 3l, q; Fig. 4b; Fig. 6h, k, u; and Fig. 7k.

12. Fig 6K should show the full native gel so that all MW specimens of PRDX1 can be seen

Response:

We thank the reviewer for this suggestion. Due to space limitations, the full uncropped images of all native gels showing all molecular weight species of PRDX1 have been provided in the Source Data file.

Reviewer #2 (Remarks to the Author):

In the presented manuscript, Yang, Zhao Peng and Lv et al. discovered KIF11 as a novel downstream target of the Norrin/ β -catenin pathway in HRECs in a screen. Mutations in KIF11 have been previously reported in the context of human eye diseases. Besides their models of endothelial cells of the eyes and mouse retinas (e.g. Fig. 9a), the authors might want to step a little further and test KIF11 in general models of ferroptosis, standard cells and assays that might expand their discovery from eye-specific to generally important to ferroptosis. I therefore suggest a couple of cell culture experiments in my list of major concerns below. Once this is performed, the manuscript will be of interest not only to eye specialists, but also to the broader readership of a journal such as Nature Communications.

In summary, at the current stage, the manuscript is quite specialized, but I can envision a broader role for their discovery in ferroptosis which I recommend to be tested in some detail.

Major concerns

1. What is the role of KIF-11 in ferroptosis, if any? This is easy to investigate by knocking out and back in AKR1 in standard ferroptosis cell lines (HT1080, Pfa1) treated with various ferroptosis inducers of all classes (erastin, RSL3, FIN56, FINO2). As readout systems, time lapse imaging, PI staining, LDH release, any other vital dye, annexin V etc should be investigated?

Response:

We thank the reviewer for this important comment.

It is indeed necessary to investigate the role of KIF11 in standard ferroptosis cell lines. Due to the unavailability of Pfa1 cells, we were unable to conduct the corresponding experiments. Instead, we have performed experiments on HRECs and HT1080 cells, which we believe provide supporting evidence for our conclusions.

In accordance with the reviewer's suggestion, we have incorporated time-lapse live-cell imaging of PI, Annexin V, as well as BODIPY staining in control and *KIF11*-knockdown HRECs and HT1080 cells. Since *KIF11* knockdown is sufficient to induce hallmarks of ferroptosis and trigger rapid cell death in HRECs and HT1080 cells, we did not apply additional ferroptosis inducers. The *KIF11*-depleted HRECs and HT1080 cells exhibited hallmark features of ferroptosis, including increased cell death and accumulated lipid peroxidation (Fig. 3p, q and Supplementary Fig. 8a-c, g-k). We have also performed LDH release assays and LC-MS/MS targeted analysis of peroxidized lipids in control and *KIF11* knockdown HRECs. As expected, the results revealed that knockdown of *KIF11* resulted in increased LDH release (Supplementary Fig. 8d) and elevated phospholipid peroxidation (Fig. 3s).

Additionally, in agreement with those observed in HRECs, *KIF11*-depleted HT1080 cells also exhibited alterations of ferroptosis-related proteins, including a prominent downregulation of GPX4, xCT, PCBP1, and SLC3A2. The related results have been incorporated in Supplementary Fig. 8e, f.

However, as efficient gene manipulation in HRECs requires viral transduction, we were unable to perform KIF11 restoration experiments in *KIF11*-knockdown cells. Also, due to the limited proliferative capacity of primary cells and the rapid inhibition of proliferation and cell death following *KIF11* knockout, we were unable to establish and

select a stable *KIF11* knockout HREC using the CRISPR-Cas9 system. Despite that, we have incorporated ferroptosis inhibitor Fer-1 treatment assays using time-lapse live-cell imaging of PI, Annexin V, as well as BODIPY staining, to assess role of β -catenin/KIF11 axis in ferroptosis. The results revealed marked attenuation of lipid peroxidation and prevention of cell death in *KIF11*- (Fig. 4l, m) or *CTNNB1*-depleted (Fig. 7t, u). Moreover, Fer-1 treatment also revealed partial rescue effects of retinal vascular development in *Kif11* cKO (Fig. 9b-d) and *Ctnnb1* cKO (Fig. 9e-g) mice. These findings establish a mechanistic link between β -catenin/KIF11 deficiency and ferroptosis.

Collectively, the revised and previous results demonstrate that KIF11 deficiency leads to ferroptosis in endothelial cells via downregulation of the xCT/GPX4 axis, independent of ferroptosis inducers.

2. What is the baseline expression and overall relevance of the reported Kif-11-system? Western blots should be provided with established (ko-controlled) antibodies. How does the absence of KIF-11 relate to the known ferroptosis-controlling systems such as GPX4, FSP1 etc.? How does it affect the baseline levels of hydroperoxides, ether lipids etc?

Response:

We thank the reviewer for highlighting this important concern. We have incorporated the data relating to baseline expression of KIF11 in mouse retinas and the disturbance of ferroptosis upon *KIF11* depletion:

1) **Baseline expression of KIF11 in mouse retinas:** As described in our response to reviewer 1's first comment, we have performed scRNA-seq on P6 retinal vascular cells isolated from wild-type, *Ctnnb1* cKO, and *Tspan12* KO mice. The resulting data were integrated with relevant public datasets from adult mice (P60 WT and Ndp KO, P26 WT and *Tspan12* KO), which revealed that *Kif11* expression is predominantly restricted to proliferative endothelial cells in postnatal retinas and is largely absent in adult retinas (P26 and P60) (Fig. 2d-g and Supplementary Fig. 2d, e). We have also performed RT-qPCR and western blot analyses using a KO-validated antibody on wild-type retinas collected at P1, P7, P14, P21, and P28. These experiments revealed prominent baseline KIF11 expression peaking in early postnatal retinas (P1 and P7) and declining rapidly thereafter (Fig. 2i and Supplementary Fig. 2g). The same KO-validated antibody confirmed the ablation of KIF11 in *KIF11* knockdown HRECs (Supplementary Fig. 4a-c), HT1080 cells (Supplementary Fig. 8e, f), and *Kif11* cKO mice lung lysates (Fig. 8f, g).

2) **The role of KIF11 in ferroptosis-regulating systems:** The previous version of this manuscript only assessed alterations in the GPX4 axis following *KIF11* ablation in

HRECs. As noted by the reviewer, we have demonstrated the disturbance of ferroptosis-related proteins within the GPX4 system in *KIF11*-depleted HT1080 cells (Supplementary Fig. 8e, f) and *Kif11* cKO mice lung lysates (Fig. 8f, g). Considering the pivotal roles of ferroptosis suppressor protein 1 (FSP1)^{7, 8} and dihydroorotate dehydrogenase (DHODH)⁹ in ferroptosis regulation, it is indeed important to examine whether these axes are also affected upon *KIF11* deletion. Western blot analysis confirmed that the expression levels of both FSP1 and DHODH remained largely unchanged upon *KIF11* knockdown in HRECs (Fig. 3i, j). The antibodies targeting ferroptosis-related proteins have been used in multiple previously published studies and validated by the respective manufacturers, including anti-DHODH¹⁰, anti-FSP1¹¹, anti-GPX4¹², anti-xCT¹³, anti-SLC3A2¹⁴, anti-ACSL4¹⁵, and Anti-HO-1¹⁶.

As mentioned by the reviewer, it is indeed essential to evaluate the baseline of ether lipids and hydroperoxides upon *KIF11* depletion. However, the insufficient number of *KIF11*-knockdown HRECs, due to compromised proliferation and activated ferroptosis, precluded direct quantification of ether lipids and hydroperoxides using untargeted LC-MS/MS. Instead, we have performed targeted LC-MS/MS analysis of ferroptosis-associated phospholipid peroxidation^{7, 17, 18, 19} and revealed that *KIF11* knockdown enhanced the peroxidation of phospholipids (phosphatidylethanolamines [PEs] and phosphatidylcholines [PCs]) and their associated ether-linked forms (Fig. 3t). Notably, GSSH is a major cellular hydroperoxide that serves as a reducing equivalent for GPX4 to detoxify lipid hydroperoxides. In *KIF11*-depleted cells, both GPX4 protein levels and enzymatic activity were significantly reduced, providing indirect evidence of impaired hydroperoxide-dependent antioxidant defense. Hydroperoxides can inhibit lipid peroxidation by scavenging reactive aldehydes such as MDA²⁰. Consistently, MDA levels were significantly increased in *KIF11*-knockdown cells, further supporting the notion of functionally reduced hydroperoxides. Taken together, these results provide indirect but converging evidence that *KIF11* depletion compromises the hydroperoxide system, contributing to increased ferroptosis susceptibility.

3. How does KIF-11 regulate ferroptosis? This requires the monitoring of cellular systems (HT1080, NIH3T3, Pfa1 and potentially the cells used in the screen in Fig. 1) over time. How plastic are the above-mentioned systems once induced in the absence of KIF-11? How does the peroxidative lipid profile assessed by LC/MS change over time? How does this phenocopy MALDI-imaging of the corresponding lipids in section of these eyes?

Response:

We thank the reviewer for highlighting this important concern. In line with the reviewer's suggestion, we have incorporated time-lapse live-cell imaging of PI, annexin V, as well as BODIPY staining in control and *KIF11*-knockdown HRECs and HT1080 cells. These assays revealed hallmark features of ferroptosis, including increased cell death and accumulated lipid peroxidation in both cells depleted with *KIF11* (Fig. 3p, q and Supplementary Fig. 8a-c, g-k). Notably, *KIF11* knockdown is sufficient to trigger rapid cell death with phospholipid peroxidation in HRECs and HT1080 cells without additional ferroptosis inducers, indicative of its susceptibility to ferroptosis (Fig. 3p, q and Supplementary Fig. 8a-c, g-k).

Furthermore, as suggested by the reviewer, we have also performed targeted LC-MS/MS analysis of ferroptosis-associated phospholipid peroxidation in control and *KIF11* knockdown HRECs. As expected, *KIF11* knockdown in HRECs led to elevated phospholipid peroxidation and their associated ether forms (Fig. 3t).

We agree that confirming the elevation of the corresponding lipid peroxidation in *Kif11* cKO mouse retinal ECs using MALDI-imaging would be valuable. However, due to technical limitations, we were unable to perform lipid peroxidation analysis using MALDI at this stage. Despite that, other evidence—including TEM of mouse retinas (Fig. 8e) and western blot analysis of mouse lung lysates (Fig. 8j, k)—confirmed the activation of ferroptosis in *Kif11* deficient mice.

Taken together, these findings strongly support the occurrence of ferroptosis upon *Kif11* depletion in mice, despite the current lack of direct MALDI-imaging evidence.

4. Fig. 7-9 are a strong part of this manuscript. But which kinase is responsible for the LLPS and PRDX1 phosphorylation, and how does KIF-11 regulate it?

Response:

We greatly appreciate the reviewer's insightful questions. A previous report identified Src as a tyrosine kinase that serves as a critical upstream regulator of PRDX1-Y194 phosphorylation²¹. Therefore, we have performed western blot analysis to assess PRDX1 phosphorylation in *KIF11*-depleted HRECs treated with the Src kinase inhibitor PP1. As anticipated, increasing concentrations of PP1 resulted in a dose-dependent reduction of PRDX1 phosphorylation (Supplementary Fig. 10i, j). Furthermore, PP1 treatment in HRECs resulted in a shift of PRDX1 aggregation from HWM complexes to LMW proteins (Fig. 5n, o). These results indicate that Src kinase activity is responsible for the phosphorylation and subsequent LLPS of PRDX1 upon *KIF11* knockdown.

To investigate how KIF11 regulates the phosphorylation of PRDX1 by Src, we have incorporated miniTurboID based co-immunoprecipitation (Co-IP) analysis to capture transient kinase-substrate interactions and examine whether KIF11 spatially occupies the Src-binding site on PRDX1, thereby preventing Src-mediated phosphorylation. The results confirmed the interaction between Src-miniTurboID-FLAG and PRDX1-HA (Fig. 5p-r). Notably, KIF11 overexpression markedly reduced the amount of biotin-labeled PRDX1, together with decreased P-PRDX1 levels in cell lysates (Fig. 5s-u), suggesting that KIF11 competes with Src for PRDX1 binding and phosphorylation. Collectively, these data suggest that KIF11 exerts an inhibitory effect on the Src/PRDX1 interaction, possibly through its spatial occupancy of the Src-binding site on PRDX1.

5. The authors conclude that autophagy dependent ferroptosis is key here. However, they do not demonstrate the requirement of the “autophagy” part. Therefore, I highly recommend to reduce this controversial term “ferroptosis” instead of “autophagy-dependent ferroptosis”. As with other regulated cell death pathways, autophagy is triggered mostly in parallel to the cell death pathways of most kinds. In that sense, autophagy is rather a counterregulatory attempt of the cell to improve its energy profile and may be interpreted as a rescue attempt, also during the process of ferroptosis.

Response:

We agree with the reviewer that the term “autophagy-dependent ferroptosis” is controversial. As suggested, we have replaced this term with either “ferroptosis” or “autophagy-accompanied ferroptosis”, and most of the autophagy-related images have been moved to the Supplementary Figures. Additionally, we have reduced the descriptions related to “autophagy” and “autophagy-dependent ferroptosis” in the main text to improve focus and clarity.

Reviewer #3 (Remarks to the Author):

The overall study has positive significance for the mechanistic research of KIF11. The authors conclude that: modulation of the Norrin pathway induces changes in KIF11; KIF11 regulates alterations in the autophagy and ferroptosis pathways; KIF11 interacts with PRDX1, and the loss of the KIF11/PRDX1 interaction leads to PRDX1 polymerization and triggers liquid-liquid phase separation (LLPS). Additionally, ferroptosis inhibitors and PRDX1 overexpression can partially rescue the vascular developmental defects caused by KIF11.

However, this study has several notable limitations:

1. Logical Inconsistencies: The experimental design is fragmented, with many proposed connections relying excessively on bioinformatic correlations. The authors have not established a complete mechanistic chain through functional experiments, and some of the bioinformatic analyses appear redundant.

Response:

We thank the reviewer for this critical comment.

(1) To improve logical consistency, we have removed or relocated redundant bioinformatic analyses to the Supplementary Figures: 1) GSEA corresponding to quadrants 4 and 6 in the nine-quadrant plot previously shown in Fig. 5d–g has been removed; 2) the volcano plot, enrichment dot plot, heatmaps, and GSEA running score plots related to the transcriptome of *KIF11* knockdown, previously shown in of Fig. 4e–k and o–u, have been moved to the Supplementary Figures to maintain the overall flow and coherence of the manuscript.

(2) In response to the reviewer's constructive feedback, we have conducted multiple new experiments over the past few months to complete a mechanistic chain. Significant advancements in our experiments include:

Incorporation of scRNA-seq data: We have incorporated scRNA-seq on retinal vascular cells of classical FEVR-associated *Ctnnb1* EC-specific conditional knockout (*Ctnnb1* cKO) and *Tspan12* global knockout (*Tspan12* KO). Integrated scRNA-seq analysis, together with multiple validation experiments, confirmed that *Kif11* is expressed specifically in proliferative (non-quiescent) endothelial cells in the postnatal mouse retina, but not in adult retinal ECs (Fig. 2d–g and Supplementary Fig. 2d, e). These findings are consistent with the observation that loss-of-function mutations in *KIF11* lead to inherited diseases such as FEVR, which typically manifest during infancy or early childhood. More importantly, scRNA-seq analysis confirmed the downregulation of *Kif11* levels in total ECs (Fig. 2d) and proliferative ECs (Fig. 2h) of both *Ctnnb1* cKO and *Tspan12* KO mice retina. Therefore, the new results incorporated in the current version further solidify the conclusion that *KIF11* is regulated by the β -catenin signaling.

Additional ferroptosis-related experiments: We have incorporated multiple experiments to further validate the relationship between the β -catenin/KIF11/PRDX1 axis and ferroptosis. In HRECs, we have performed time-lapse live-cell imaging with PI, annexin V (Supplementary Fig. 8a–c), and BODIPY staining (Fig. 3o, p), along with LDH release assays (Supplementary Fig. 8d) and LC-MS /MS targeted analysis of phospholipid peroxidation (Fig. 3t). These results support the conclusion that loss of

KIF11 function is associated with ferroptosis in HRECs. To assess whether KIF11 plays a broader role in ferroptosis beyond endothelial cells, we conducted additional experiments in HT1080 cells, a model system commonly used for ferroptosis studies. Time-lapse live-cell imaging with PI, annexin V, and BODIPY staining (Supplementary Fig. 8g-k), together with western blot analysis of core ferroptosis-related proteins (Supplementary Fig. 8e, f), yielded results consistent with those observed in HRECs, indicating that KIF11 contributes to the regulation of ferroptosis across multiple cell types.

Additional function experiments to validate the mechanistic role of KIF11 on the phosphorylation of PRDX1: Our new results show that PRDX1 phosphorylation in *KIF11*-depleted HRECs is primarily mediated by Src, as evidenced by dose-dependent inhibition with PP1 (Supplementary Fig. 10i, j). Additionally, we have performed miniTurboID-based Co-IP experiments to capture transient interaction between Src and PRDX1 (Fig. 5p-u), which indicate that KIF11 inhibits the Src/PRDX1 interaction, likely through spatial occupancy of the Src-binding site on PRDX1, thereby preventing Src-mediated phosphorylation.

Additional *in vitro* and *in vivo* rescue experiments: To further address the role of the β -catenin/KIF11/PRDX1 axis in ferroptosis, we have performed ferroptosis inhibitor (Fer-1) treatment assays in HRECs, combined with time-lapse live-cell imaging using PI, annexin V, and BODIPY staining. The results demonstrated a marked attenuation of lipid peroxidation and prevention of cell death in HRECs depleted of *KIF11* (Fig. 4l, m) or *CTNNB1* (Fig. 7t, u). We have also conducted rescue experiments *in vivo*, in which Fer-1 treatment partially alleviated the vascular defects observed in *Ctnnb1* cKO retinas (Fig. 9e-g). Furthermore, retro-orbital injection of KIF11-expressing lentivirus into *Ctnnb1* cKO mice similarly resulted in partial restoration of retinal vascular progression (Fig. 2p-r).

Collectively, our results demonstrate that loss of KIF11 function in endothelial cells led to autophagy-accompanied ferroptosis by releasing its competitive role on the Src-mediated phosphorylation and triggering liquid-liquid phase separation (LLPS) of PRDX1, ultimately leading to defective vascularization in FEVR.

2. Weak Evidence for Autophagy-Dependent Ferroptosis: Although KIF11 is shown to induce autophagy and ferroptosis separately, there is insufficient experimental evidence to definitively demonstrate that KIF11 triggers ferroptosis through autophagy-dependent mechanisms.

Response:

We thank the reviewer for raising this concern. As also suggested by Reviewer 2, we replaced the term “autophagy-dependent ferroptosis” with either “ferroptosis” or “autophagy-accompanied ferroptosis”, and most of the autophagy-related images have been moved to the Supplementary Figures. Additionally, we have reduced autophagy-related descriptions in the main text to improve focus and clarity.

3. The reliance on a single shRNA for knockdown experiments undermines the reliability of these findings.

Response:

We thank the reviewer for the insightful suggestion. As was also suggested by Reviewer 1, we have incorporated a second shRNA targeting another site for the knockdown of *CTNNB1*, *LRP5*, *FZD4*, or *TSPAN12* to confirm the downregulation of KIF11 in HRECs; the related statistical analyses have been incorporated in the **Supplementary Fig. 1e-h**. Additionally, a second shRNA targeting another site for *KIF11* or *PRDX1* was performed to validate consistency by western blot analyses concerning ferroptosis-related proteins. The related images and statistical analyses for *KIF11*- and *PRDX1*-depletion have been incorporated in the **Supplementary Fig. 7h-j** and **Supplementary Fig. 12h-j**, respectively.

4. Abrupt Presentation of KIF11 Mutation Data: The inclusion of KIF11 mutation results lacks contextual integration with the overall narrative, appearing disconnected from the core mechanistic exploration.

Response:

We thank the reviewer for raising this concern. We agree with the reviewer that the identification of novel frameshift or nonsense *KIF11* mutations is less central to the mechanistic focus of this article. Accordingly, the pedigree and clinical images have been moved to **Supplementary Fig. 3a-i**. However, the western blot results for the expression of the mutated KIF11 proteins have been retained in **Fig. 2o**, as these mutations were tested for their interactions with PRDX1 in **Fig. 4q**. We believe this inclusion is important for establishing a clinical link between the disrupted PRDX1/KIF11 interaction and FEVR.

5. Incomplete Animal Experiments: The methodology for drug treatment in animal models is inadequately described. Phenotypic analysis is superficial, focusing primarily on vascular developmental status while neglecting critical assessments (e.g., blood-retinal barrier integrity).

Response:

We thank the reviewer for highlighting this issue and apologize for the previous omission of the drug treatment methodology in animal models. We have carefully reviewed the article and added the missing description to the Method section.

Regarding the phenotypic analysis, we agree that assessing blood-retinal barrier (BRB) integrity is pivotal for vascular research. Accordingly, we have incorporated Ter119 staining of erythrocytes, which reflects transcellular BRB integrity, and Sulfo-biotin, a small molecular tracer indicating paracellular BRB integrity, to evaluate the BRB in the presence or absence of KIF11. However, the results revealed no obvious leakage of either signal, indicating undisturbed vascular integrity in *Kif11* cKO retinas (Supplementary Fig. 13b, c).

6. Given the close relationship between vascular development and overall growth in mice, the authors should provide body weight statistics for all experimental groups in the supplementary materials.

Response:

We thank the reviewer for raising this concern and have incorporated the body weight statistics in the Supplementary Materials. Neither EC-specific depletion of *Ctnnb1* or *Kif11*, nor retro-orbital lentivirus or intraperitoneal Fer-1 injection, affected mouse body weight. Specifically, the body weight for P7 wild-type and *Ctnnb1* cKO treated with control lentivirus, and *Ctnnb1* cKO treated KIF11-expressing lentivirus have been incorporated in Supplementary Fig. 3k. The body weight for P7 wild-type and *Kif11* cKO mice have been incorporated in Supplementary Fig. 13a. The body weight for P7 wild-type and *Kif11* cKO treated with DMSO, and *Kif11* cKO treated Fer-1 have been incorporated in Supplementary Fig. 14a. The body weight for P7 wild-type and *Ctnnb1* cKO treated with DMSO, and *Ctnnb1* cKO treated Fer-1 have been incorporated in Supplementary Fig. 14b. The body weight for P7 *Kif11* cKO treated with control, PRDX1-WT, and PRDX1-mut lentivirus have been incorporated in Supplementary Fig. 14c.

References

1. Mukerjee A, Shankardas J, Ranjan AP, Vishwanatha JK. Efficient nanoparticle mediated sustained RNA interference in human primary endothelial cells. *Nanotechnology* **22**, 445101 (2011).
2. Levey J, et al. The MDM2-p53 axis regulates norrin/frizzled4 signaling and blood-CNS barrier function. *Sci Signal* **18**, eadt0983 (2025).

3. Ding J, *et al.* Therapeutic blood-brain barrier modulation and stroke treatment by a bioengineered FZD(4)-selective WNT surrogate in mice. *Nat Commun* **14**, 2947 (2023).
4. Zhang L, *et al.* A Frizzled4-LRP5 agonist promotes blood-retina barrier function by inducing a Norrin-like transcriptional response. *iScience* **26**, 107415 (2023).
5. Rocha-Ferreira E, *et al.* A Neonatal Rodent Model of Retroorbital Vein Injection. *J Vis Exp*, (2024).
6. Li S, *et al.* Retro-orbital injection of FITC-dextran is an effective and economical method for observing mouse retinal vessels. *Mol Vis* **17**, 3566-3573 (2011).
7. Doll S, *et al.* FSP1 is a glutathione-independent ferroptosis suppressor. *Nature* **575**, 693-698 (2019).
8. Bersuker K, *et al.* The CoQ oxidoreductase FSP1 acts parallel to GPX4 to inhibit ferroptosis. *Nature* **575**, 688-692 (2019).
9. Mao C, *et al.* DHODH-mediated ferroptosis defence is a targetable vulnerability in cancer. *Nature* **593**, 586-590 (2021).
10. Qiu S, *et al.* Mitochondria-localized cGAS suppresses ferroptosis to promote cancer progression. *Cell Res* **33**, 299-311 (2023).
11. Shan X, *et al.* Targeting ferroptosis by poly(acrylic) acid coated Mn(3)O(4) nanoparticles alleviates acute liver injury. *Nat Commun* **14**, 7598 (2023).
12. Hu JJ, *et al.* Time-restricted feeding protects against septic liver injury by reshaping gut microbiota and metabolite 3-hydroxybutyrate. *Gut Microbes* **17**, 2486515 (2025).
13. Nhung DT, Yousif OEA, Kwon B. Sorafenib induces ferroptosis in human renal cell carcinoma cells through CCAT/enhancer-binding protein homologous protein. *Biochem Biophys Res* **43**, 102143 (2025).
14. Zhan D, *et al.* Targeting Caveolin-1 in Multiple Myeloma Cells Enhances Chemotherapy and Natural Killer Cell-Mediated Immunotherapy. *Adv Sci (Weinh)* **12**, e2408373 (2025).
15. Wang F, *et al.* BNC1 deficiency-triggered ferroptosis through the NF2-YAP pathway induces primary ovarian insufficiency. *Nat Commun* **13**, 5871 (2022).
16. Li H, Wang J, Jin Y, Lin J, Gong L, Xu Y. Hypoxia upregulates the expression of lncRNA H19 in non-small cell lung cancer cells and induces drug resistance. *Transl Cancer Res* **11**, 2876-2886 (2022).

17. Mo C, *et al.* Dopaminylation of endothelial TPI1 suppresses ferroptotic angiocrine signals to promote lung regeneration over fibrosis. *Cell Metab* **36**, 1839-1857 e1812 (2024).
18. Slatter DA, *et al.* Mapping the Human Platelet Lipidome Reveals Cytosolic Phospholipase A2 as a Regulator of Mitochondrial Bioenergetics during Activation. *Cell Metab* **23**, 930-944 (2016).
19. Aldrovandi M, *et al.* Specific oxygenation of plasma membrane phospholipids by *Pseudomonas aeruginosa* lipoxygenase induces structural and functional alterations in mammalian cells. *Biochim Biophys Acta Mol Cell Biol Lipids* **1863**, 152-164 (2018).
20. Barayeu U, *et al.* Hydropersulfides inhibit lipid peroxidation and ferroptosis by scavenging radicals. *Nat Chem Biol* **19**, 28-37 (2023).
21. Liu J, *et al.* The E3 Ligase TRIM16 Is a Key Suppressor of Pathological Cardiac Hypertrophy. *Circ Res* **130**, 1586-1600 (2022).

Response to Reviewer's Comments

General response to reviewers' comments:

We sincerely appreciate the reviewers' positive feedback and are grateful for the constructive suggestions during the previous round of revision, which have significantly improved our manuscript.

We have carefully considered the issues raised and have addressed each one comprehensively in the revised manuscript. Detailed explanations of the changes made can be found in our point-by-point responses.

REVIEWER COMMENTS

Reviewer #1 (Remarks to the Author):

The authors conducted additional experiments that strengthen the conclusions of the study. With respect to the claim that KIF11 is a target gene of norrin signaling, the authors focussed on proliferating ECs and found a reduction of KIF11 mRNA in P6 proliferating ECs in Tspan12 and Ctnnb1 mutant cells (Fig. 2h, statistics not described in the figure legend). Lentivirus mediated delivery of KIF11 partially rescued retinal vascular outgrowth in Ctnnb1 mutant retinas. Whether norrin induces KIF11 in HRMVECs remains unclear. Together, concerns raised by this reviewer were sufficiently addressed.

Response:

We appreciate the reviewer's thoughtful comments. We have incorporated the statistical method (Unpaired student's t-test) for Fig. 2h in the figure legend.

According to reviewer's suggestion, we have incorporated RT-qPCR analysis of *KIF11* in the presence or absence of exogenous Norrin in HRECs (Supplementary Fig. 1i), which demonstrated that exogenous Norrin significantly increased *KIF11* expression in HRECs. We have added this information in the revised manuscript page 5.

Reviewer #2 (Remarks to the Author):

The authors have greatly improved their manuscript. However, I recommend to test one more important control in the HT1080 cells (Supplementary Figure 8). Would the cell death observed upon knockdown of KIF11 be reversed by the addition of a ferroptosis inhibitor (e.g. Fer-1)? If the authors cannot provide a crCas9-mediated knockout of Kif-11 in HT1080 cells, they should at least discuss their limitations (PMID38489367). Otherwise, I recommend this manuscript for publication!

Response:

We thank the reviewer for these important comments. It is indeed necessary to investigate whether a ferroptosis inhibitor could reverse the cell death observed in *KIF11* KD HT1080 cells. Thus, in accordance with the reviewer's suggestion, we have performed live-cell imaging of PI, Annexin V, and BODIPY staining in *KIF11* KD HT1080 cells treated with ferroptosis inhibitor Fer-1 or vehicle. The results revealed marked prevention of lipid peroxidation and cell death in *KIF11*-depleted HT1080 cells after treated with ferroptosis inhibitor Fer-1 (Supplementary Fig. 8l, m). The related descriptions have added in the revised manuscript page 11.

Furthermore, we agree with the reviewer that, in light of recent findings showing that siRNA treatment can significantly sensitize cells to ferroptosis independently of target protein knockdown¹, generating stable *KIF11* knockout cell lines using the CRISPR-Cas9 system in HRECs and HT1080 cells would be highly informative. However, we were unable to generate the stable *KIF11*-knockout HRECs or HT1080 cell lines owing to the rapid proliferation arrest and increased cell death upon *KIF11* depletion. Nevertheless, in this study, a lentiviral vector expressing a non-targeting shRNA was used as a negative control (the "CTRL" group in the figures) to minimize potential confounding effects of RNAi treatment. Crucially, this control exhibited no detectable effect on lipid peroxidation or cell death in HRECs and HT1080 cells. We have revised the manuscript to acknowledge this limitation, which is now addressed in the Discussion section (Page 19).

Reviewer #3 (Remarks to the Author):

The manuscript has significantly improved in this revised version. Both the experimental design and the writing are more logically structured than before. The authors have incorporated an additional shRNA to assess their key conclusion, and they have reorganized the section discussing the relationship between *KIF11* and autophagy in conjunction with ferroptosis. Moreover, the mechanisms underlying how the LoF of *KIF11* impairs endothelial cells through interaction with *PRDX1* in the context of FEVR are now well addressed. However, there are several minor issues that still require refinement.

1. Lines 246-247: The authors conclude, "Taken together, these results suggest that the loss of *KIF11* might trigger autophagy during retinal vascular development." However, in the in vitro experiments and bioinformatics analysis involving Primary Human Retinal Microvascular Endothelial Cells, the donor's age is not provided on the official website of the primary cell line. Therefore, it is imprecise to attribute this phenotype to "retinal vascular development." This conclusion would be more appropriate in the context of the in vivo results.

Response:

We thank the reviewer for the valuable suggestions. In response, we have removed the phrase "during retinal vascular development" and revised the statement as follows: "Taken together, these results suggest that the loss of *KIF11* might trigger autophagy in HRECs".

2. Lines 278-279: The statement regarding the protective effect of *KIF11* against ferroptosis is inaccurate, as no experimental data regarding the overexpression of *KIF11* to mitigate ferroptosis are presented in this section.

Response:

We sincerely appreciate the reviewer's constructive comments. Accordingly, we have revised this statement as follows: "These results imply that *KIF11* depletion increases ferroptotic vulnerability in HRECs".

3. The MWs of *PRDX1*'s LMW and HWM forms should be mentioned, or at least an approximate scale should be provided.

Response:

We thank the reviewer for the valuable suggestions. According to previous reports, PRDX1 monomers undergo oxidation to form peroxidase-inactive homodimers, a transformation facilitates assembly into chaperone-active polymers and trigger PRDX1 LLPS^{2, 3, 4, 5}. The HMW and LMW PRDX1 in our non-reducing PAGE analysis are peroxidase-inactive dimeric and peroxidase-active monomeric forms, respectively. We have included the approximate molecular weights of PRDX1's LMW and HWM forms in Fig. 4x, Fig. 5g, Fig. 5j, Fig. 5n, Fig. 7i, and Supplementary Fig. 10e.

References

1. von Massenhausen A, *et al.* Treatment with siRNAs is commonly associated with GPX4 up-regulation and target knockdown-independent sensitization to ferroptosis. *Sci Adv* **10**, eadk7329 (2024).
2. Cao J, *et al.* Prdx1 inhibits tumorigenesis via regulating PTEN/AKT activity. *EMBO J* **28**, 1505-1517 (2009).
3. Jang HH, *et al.* Phosphorylation and concomitant structural changes in human 2-Cys peroxiredoxin isotype I differentially regulate its peroxidase and molecular chaperone functions. *FEBS Lett* **580**, 351-355 (2006).
4. Jang HH, *et al.* Two enzymes in one; two yeast peroxiredoxins display oxidative stress-dependent switching from a peroxidase to a molecular chaperone function. *Cell* **117**, 625-635 (2004).
5. Jiang Y, *et al.* Phosphorylation-Regulated Dynamic Phase Separation of HIP-55 Protects Against Heart Failure. *Circulation* **150**, 938-951 (2024).

Response to Reviewer's Comments

REVIEWER COMMENTS

Reviewer #2 (Remarks to the Author):

The authors have adequately responded to all concerns raised by this referee, but still, I believe that a crCas9-based knockout is required for the conclusions. An shRNA or an siRNA in both cases may affect ferroptosis. However, this is discussed as a limitation of this work, so I recommend to publish the manuscript.

Response:

We sincerely thank the referee's positive recommendation and the constructive final feedback. As noted, we have explicitly discussed this as a limitation in the manuscript. We appreciate the referee's professional recognition of our efforts to ensure a transparent discussion of these methodologies.